# A dedicated diribonuclease resolves a key bottleneck for the terminal step of RNA degradation

Soo-Kyoung Kim[1†], Justin D Lormand[2†], Cordelia A Weiss[1†], Karin A Eger[2], Husan Turdiev[1], Asan Turdiev[1], Wade C Winkler[1]*, Holger Sondermann[2]*, Vincent T Lee[1]*

[1]Department of Cell Biology and Molecular Genetics, University of Maryland, College Park, United States; [2]Department of Molecular Medicine, College of Veterinary Medicine, Cornell University, Ithaca, United States

**Abstract** Degradation of RNA polymers, an ubiquitous process in all cells, is catalyzed by specific subsets of endo- and exoribonucleases that together recycle RNA fragments into nucleotide monophosphate. In γ-proteobacteria, 3-'5' exoribonucleases comprise up to eight distinct enzymes. Among them, Oligoribonuclease (Orn) is unique as its activity is required for clearing short RNA fragments, which is important for cellular fitness. However, the molecular basis of Orn's unique cellular function remained unclear. Here, we show that Orn exhibits exquisite substrate preference for diribonucleotides. Crystal structures of substrate-bound Orn reveal an active site optimized for diribonucleotides. While other cellular RNases process oligoribonucleotides down to diribonucleotide entities, Orn is the one and only diribonuclease that completes the terminal step of RNA degradation. Together, our studies indicate RNA degradation as a step-wise process with a dedicated enzyme for the clearance of a specific intermediate pool, diribonucleotides, that affects cellular physiology and viability.

*For correspondence:
wwinkler@umd.edu (WCW);
hs293@cornell.edu (HS);
vtlee@umd.edu (VTL)

†These authors contributed equally to this work

**Competing interests:** The authors declare that no competing interests exist.

## Introduction

Degradation of RNA is initiated by endonuclease-catalyzed cleavages; the resulting oligoribonucleotide fragments are hydrolyzed to completion by a mixture of exoribonucleases for the maintenance of cellular nucleotide pools (*Hui et al., 2014*). Unlike the conserved machineries for the synthesis of macromolecules, distinct sets of RNases are used by different organisms to degrade these oligoribonucleotides. *Escherichia coli* genomes encode eight 3′−5′ exoribonucleases, namely polynucleotide phosphorylase, RNase II, D, BN, T, PH, R and oligoribonuclease (Orn) (*Hui et al., 2014*; *Bandyra and Luisi, 2013*). A subset of these enzymes recognizes structural features of the RNA substrate, such as in tRNA, while others act on unstructured polymers (*Li and Deutscher, 1996*).

As an exoribonuclease, Orn is unique for two reasons. First, *orn* is required for viability in many γ-proteobacteria, including *E. coli* (*Ghosh and Deutscher, 1999*) and other organisms (*Palace et al., 2014*; *Kamp et al., 2013*), unlike all other known 3′−5′ exoribonucleases. This indicates that not all exoribonucleases have redundant functions despite acting on the 3′ end of RNA substrates and in several cases, including RNase R and RNase II, sharing an activity toward oligoribonucleotides (*Zuo and Deutscher, 2002*; *Cheng and Deutscher, 2002*; *Cheng and Deutscher, 2005*; *Frazão et al., 2006*). Hence, Orn appears to catalyze a particularly important step in RNA turnover. Second, Orn is a key enzyme in bacterial cyclic-di-GMP (c-di-GMP) signaling. The nucleotide second messenger c-di-GMP is produced in bacteria in response to environmental cues and controls a wide range of cellular pathways, including cell adhesion, biofilm formation, and virulence (*Krasteva and Sondermann, 2017*; *Römling et al., 2013*; *Hengge, 2009*; *Jenal et al., 2017*; *Hall and Lee, 2018*).

Since the discovery of c-di-GMP over 30 years ago, it has been known that the signal is degraded by a two-step process with a linear pGpG diribonucleotide intermediate (*Ross et al., 1987*). While the enzyme for linearizing c-di-GMP to pGpG was discovered early on (*Tal et al., 1998*; *Galperin et al., 1999*), the identity of the enzyme that degrades pGpG remained elusive. Two recent studies showed that Orn is the primary enzyme that degrades pGpG in *Pseudomonas aeruginosa* (*Cohen et al., 2015*; *Orr et al., 2015*). In an *orn* deletion strain, the accrual of linear pGpG has a profound effect on cells. Specifically, the increase in pGpG inhibits upstream phosphodiesterases that degrade c-di-GMP, thereby triggering phenotypes associated with high cellular c-di-GMP levels. However, the molecular basis of Orn's unique cellular functions in γ-proteobacteria that distinguishes it from all other exoribonucleases remains unexplained.

Since its discovery over 50 years ago (*Stevens and Niyogi, 1967*), Orn has been presumed to degrade oligoribonucleotides (*Niyogi and Datta, 1975*; *Datta and Niyogi, 1975*; *Yu and Deutscher, 1995*). This notion largely derived from assays utilizing two types of substrates. In one series of experiments, $^3$H polyuridine (poly(U)) was incubated with Orn, other enzymes, or lysates and analyzed by paper chromatography, which offers limited resolution overall (*Niyogi and Datta, 1975*; *Datta and Niyogi, 1975*; *Yu and Deutscher, 1995*). In a second set of experiments, Orn was incubated with oligoribonucleotides that had been tagged at their 5' terminus by a large fluorophore. The products of this reaction were resolved by denaturing polyacrylamide gel electrophoresis, thereby allowing for detection of products with one or more nucleotides removed from the 3' terminus (*Cohen et al., 2015*; *Mechold et al., 2006*; *Mechold et al., 2007*; *Fang et al., 2009*; *Liu et al., 2012*). In both instances, it was concluded that Orn could processively degrade 'short' oligoribonucleotides. Yet, it was not clear how Orn might selectively target 'short' RNAs, rather than simply binding to the penultimate sequence at the 3' termini of single-stranded RNA of any length.

Here, we sought to rigorously examine Orn's substrate preferences, revealing its unique properties. To that end, we incubated Orn with 5' $^{32}$P end-labeled RNAs of varying lengths and analyzed the products of that reaction over time. Our data show that Orn exhibits a surprisingly narrow substrate preference for diribonucleotides. This finding is in stark contrast to the previous studies that described a broader substrate length range and that originally gave oligoribonuclease its name. We sought to understand this remarkable substrate selectivity of Orn by determining the crystal structures of Orn in complex with pGpG and other linear diribonucleotides. These data reveal a structural basis for the diribonucleotide preference and identify key residues for recognizing diribonucleotides. Furthermore, we find that Orn is the only diribonuclease in *P. aeruginosa*. Additionally, we show that other diribonucleotides, in addition to pGpG, can affect bacterial physiology. From this we propose a general model of RNA degradation, wherein a combination of exoribonucleases process oligoribonucleotides down to diribonucleotides and Orn completes RNA recycling by cleaving diribonucleotides to nucleoside monophosphates. In this way, Orn occupies an important but discrete step in the overall RNA degradation pathway.

## Results and discussion

### Orn functions as a diribonuclease in vitro

To understand the length preference of Orn, recombinant affinity-tagged *Vibrio cholerae* Orn (Orn$_{Vc}$) was purified and tested biochemically. We used an established ligand-binding assay (*Roelofs et al., 2011*) to determine the relative substrate affinities of Orn to 5'-radiolabeled oligoribonucleotides that ranged from two to seven nucleotides in length. Unlike fluorophore-labeled RNA used in several earlier studies (*Cohen et al., 2015*; *Mechold et al., 2006*; *Mechold et al., 2007*; *Fang et al., 2009*; *Liu et al., 2012*), radiolabeling with $^{32}$P ensures that the substrate structure is unperturbed compared to native ligands. The assays were also performed in buffer lacking divalent cations to prevent complication due to enzyme activity. When spotted on nitrocellulose membranes, protein and protein-substrate complexes are sequestered at the point of application to the membrane, whereas unbound substrate diffuses due to radial capillary action (*Roelofs et al., 2011*). By this assay, quantification of the diffusion zone reveals that Orn$_{Vc}$ exhibits the highest affinity for diribonucleotide ($K_d$ $^{pGpG}$ = 90 -/+9 nM), as compared to oligoribonucleotides of greater lengths (*Table 1*) (*Orr et al., 2015*). The observed affinity of Orn$_{Vc}$ with the purification tag removed ($K_d$ $^{pGpG}$ = 80 -/+9 nM) was similar indicating the tags did not alter the

**Table 1.** Quantitative measurement of length-dependent oligoribonucleotide affinities.

| Substrate[a] | Dissociation constant $K_d$ (nM)[b] |
| --- | --- |
| GG | 90 -/+9 |
| AGG | 630 -/+80 |
| AAGG | 890 -/+90 |
| AAAGG | 2,560 -/+170 |
| AAAAGG | 3,830 -/+370 |
| AAAAAGG | 3,750 -/+530 |

[a]: RNAs were labeled with $^{32}$P at their 5' end. [b]: Affinities were measured using an established binding assay for radiolabeled RNAs (*Roelofs et al., 2011*; *Patel et al., 2014*).

interactions (*Figure 3—figure supplement 1*). Increase in the length to three or four residues reduces the affinity 7- or 10-fold, respectively (*Table 1*). Substrates with five or more bases show a greater than 28-fold reduction in affinity compared to diribonucleotides. These results suggest that Orn has a strong preference for diribonucleotides over longer oligonucleotides.

To understand whether the affinity preference reflects nuclease activity with natural substrates that are unmodified at the 5' end, we incubated Orn$_{Vc}$ with 5'-$^{32}$P-radiolabeled oligoribonucleotides of varying lengths in the presence of divalent cations that support catalysis. The products of these reactions were resolved by urea-denaturing 20% polyacrylamide gel electrophoresis (PAGE). Under these electrophoresis conditions, the mononucleotides and oligonucleotides that were tested (between 2 and 7 nucleotides in length) can be resolved. The diribonucleotide substrate in this experiment was pGpG (GG), whereas the longer oligoribonucleotides included an increasing number of adenine nucleotides at the 5' end. This arrangement ensured that the same GG sequence was maintained at the 3' end while also avoiding stable G quadruplex formation that may be observed with RNAs containing stretches of poly-G (*Kwok and Merrick, 2017*). At substrate concentrations that far exceed enzyme concentration (200:1), the diribonucleotide substrate was already fully processed to nucleoside monophosphates by 30 min (*Figure 1A*) (*Orr et al., 2015*). In contrast, longer substrates (i.e. from 3-mers to 7-mers) were not processed at their 3' end, even at 30 min (*Figure 1A and B*). These results indicate that the length preference for exoribonuclease activity of Orn on diribonucleotides is even greater than mere differences in binding affinities. This strong substrate preference stands in stark contrast to previous studies arguing Orn acts as a general exoribonuclease that cleaves oligoribonucleotides with two to seven residues in length (*Cohen et al., 2015*; *Niyogi and Datta, 1975*; *Datta and Niyogi, 1975*; *Mechold et al., 2006*; *Mechold et al., 2007*).

To determine if Orn can indeed cleave substrates longer than a diribonucleotide, the enzyme was incubated with RNA substrates at a 1:1 molar ratio. Under these conditions, the diribonucleotide substrate was completely processed to nucleoside monophosphates by the earliest time point, 20 s (*Figure 1C and D*). Orn$_{Vc}$ also facilitated the degradation of the longer RNA substrates, but only after significantly longer incubation times. For example, it required 10 min and 30 min to fully degrade 3-mer and 4-mer RNAs, respectively (*Figure 1C and D*). Cleavage was reduced further for longer RNAs; only 40–53% of the 5-mer, 6-mer and 7-mer RNAs were processed to nucleoside monophosphates at 30 min (*Figure 1C and D*), correlating with the weak affinities determined for these substrates (*Table 1*). Nonetheless, cleavage of the 7-mer required enzymatic activity since catalytically inactive Orn variants were unable to degrade the 7-mer (*Figure 1—figure supplement 1*). Of note, we observed a non-uniform distribution of degradation products for the longer RNA substrates. Specifically, the diribonucleotide intermediate was never observed as a reaction intermediate for the longer RNAs. This reaction pattern indicates that RNAs of more than two residues could accumulate, but that diribonucleotide RNAs were always rapidly processed to nucleoside monophosphates. Together, these results indicate that Orn exhibits a strong substrate preference for diribonucleotides over longer oligoribonucleotides – far greater than reported previously (*Cohen et al., 2015*; *Mechold et al., 2006*). While assays here were performed at near-physiological ionic strength, prior studies often utilized buffer lacking salt (NaCl/KCl), which may at least in part contribute to the apparent differences in catalytic activity (*Niyogi and Datta, 1975*; *Ghosh and Deutscher, 1999*; *Cohen et al., 2015*; *Mechold et al., 2006*).

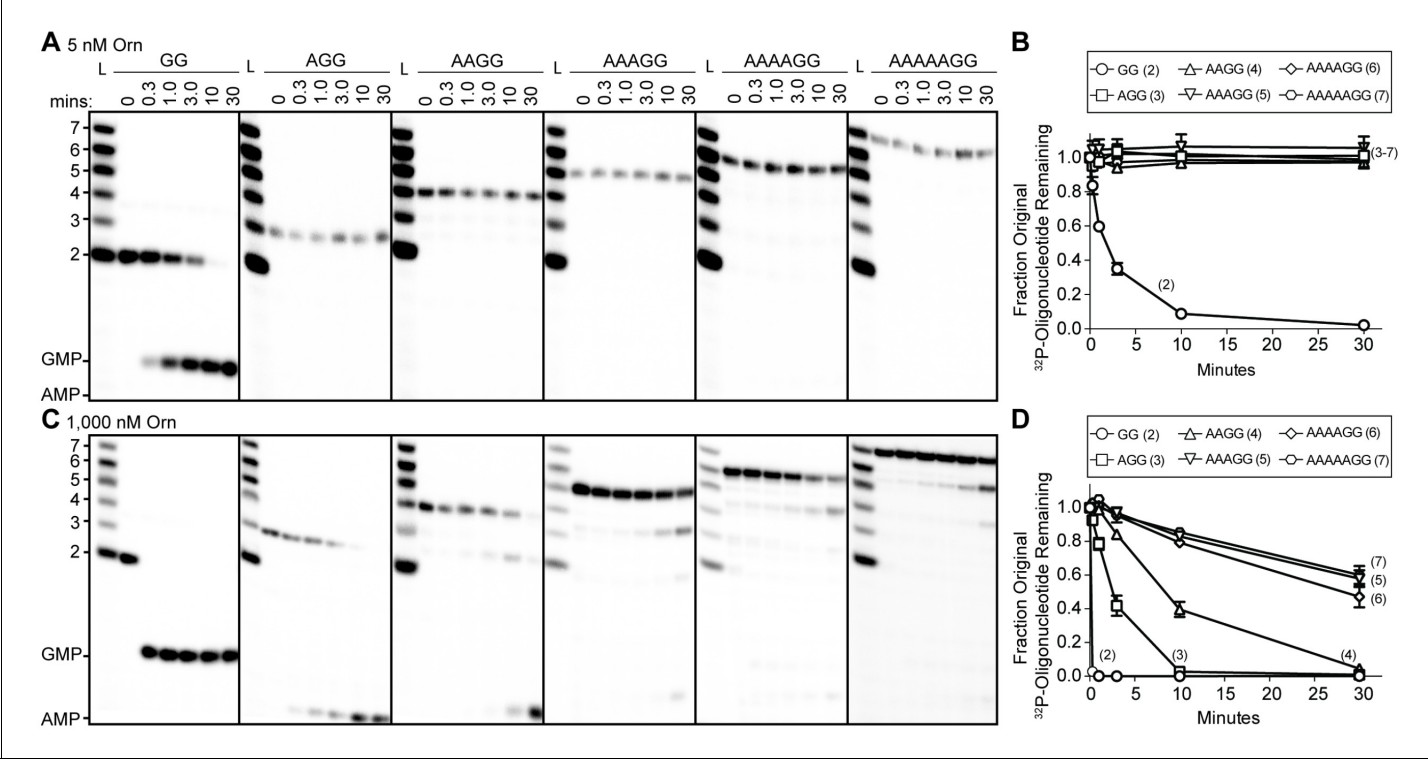

**Figure 1.** Orn has a stark preference for diribonucleotide cleavage in vitro. RNA nucleotides two to seven residues in length (1 µM, containing the corresponding $^{32}$P-labeled RNA tracer) were each subjected to cleavage over time with 5 nM (**A, B**) or 1000 nM Orn$_{Vc}$ (**C, D**). Aliquots of each reaction were stopped at indicated times (min), and assessed by denaturing 20% PAGE (**A, C**). Quantification of the intensities of bands corresponding to the amount of uncleaved initial oligonucleotide over time are plotted as the average and SD of three independent experiments (**B, D**).

The online version of this article includes the following source data and figure supplement(s) for figure 1:

**Source data 1.** Numerical data for *Figure 1B and D*.

**Figure supplement 1.** Catalytic residues of Orn$_{Vc}$ are required for degradation for $^{32}$P-AAAAAGG in vitro.

## Structure of Orn complexes with pGpG and other linear diribonucleotides

To elucidate the molecular basis for these unique properties of Orn, we set out to gain a deeper understanding of the enzyme's substrate specificity by determining Orn/substrate co-crystal structures. We initially determined the crystal structure of *P. aeruginosa* Orn (Orn$_{Pa}$) with diribonucleotide substrate; however, crystal packing contacts prevented substrate binding (data not shown). Instead, we were able to crystallize two representative homologs, Orn$_{Vc}$ (*Kamp et al., 2013*) and the human REXO2 (also known as small fragment nuclease or Sfn [*Bruni et al., 2013*]), bound to the diribonucleotide pGpG (*Figure 2A and B*; *Figure 2—source data 1*). Both proteins purified free of divalent cation, which was sufficient to prevent catalysis in crystals. Superimposing the two structures indicates their identical fold (rmsd of 0.75 Å for the protomer) (*Figure 2—figure supplement 1A*), which is preserved in the substrate-free Orn structure determined previously (rmsd of 0.54 Å for the protomer) (*Figure 2—figure supplement 1A*) (*Chin et al., 2006*). The pGpG-bound structures reveal a narrow active site that is lined by the conserved acidic residues of the signature DEDD motif (D[12], E[14], and D[112] of Orn$_{Vc}$; D[15], E[17], and D[115] of REXO2) and the general base H[158] or H[162] in Orn$_{Vc}$ or REXO2, respectively (*Figure 2A and B* and *Figure 2—figure supplement 1B* and *Figure 2—source data 2*). In Orn$_{Vc}$, the bases of the diribonucleotide buttress against aromatic residues W[61] and Y[129], the latter being contributed from the second half-side of the dimeric enzyme. The corresponding residues, W[64] and Y[132], are conserved in REXO2. Residue L[18] in Orn$_{Vc}$ (L[21] in REXO2) wedges in-between the two bases. Most notably, residues S[108], R[130], S[135] and the hydroxyl group on Y[129] (S[111], R[133], S[138], and Y[132] in REXO2) form hydrogen bonds with the 5' phosphate of pGpG, capping the

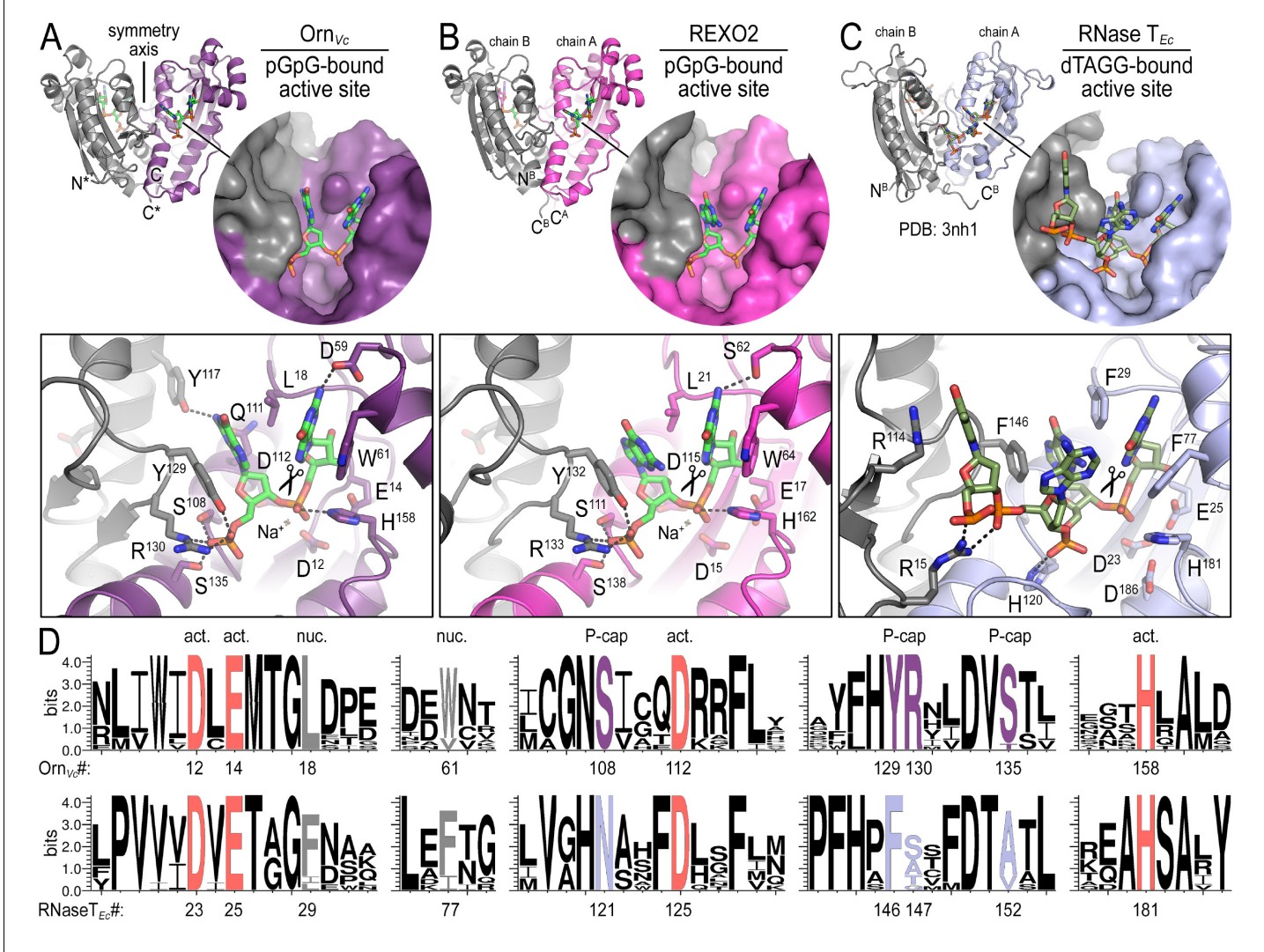

**Figure 2.** Structures reveal Orn's conserved substrate preference for diribonucleotides. Crystal structures of pGpG-bound *V. cholerae* Orn (**A**) and human REXO2 (**B**) are shown in comparison to *E. coli* RNase T bound to substrate (PDB 3nh1; *Hsiao et al., 2011*) (**C**), another DnaQ-fold 3'−5' exoribonuclease with a DEDD(h) active site motif. The top panels show ribbon representations of the dimeric enzymes. The insets are surface representations of the enzymes' active sites shown in similar orientations. The bottom panel describes the active site residues involved in RNA binding and catalysis. Residue numbering for REXO2 refers to its cytosolic isoform lacking the mitochondria-targeting pre-sequence. The sequence logos in (**D**) were constructed based on multi-sequence alignments of Orn and RNase T orthologs. Overall sequence identity ranges from 43 to 70% for Orn and 46 to 69% for RNase T. Sequence identifiers are provided in *Figure 2—source data 2*. Sequence logos were plotted using WebLogo (*Crooks et al., 2004*). Conserved residues of the active site's DEDD motif ('act.'; red), for ribonucleotide base binding ('nuc.'; gray), and of the phosphate cap ('P-cap'; purple) are highlighted.

The online version of this article includes the following source data and figure supplement(s) for figure 2:

**Source data 1.** Data collection and refinement statistics.
**Source data 2.** Sequences used to calculate surface conservation and generate Weblogos.
**Figure supplement 1.** Structural comparison of Orn orthologs, RNase T and ExoI.
**Figure supplement 2.** Structural comparison of Orn$_{Vc}$ and REXO2 bound to diverse diribonucleotides.

substrate. This phosphate cap creates a major constriction of the active site, which is not observed in structurally related 3'−5' exoribonucleases such as RNase T or ExoI (*Figure 2C* and *Figure 2—figure supplement 1* and *Figure 2—source data 2*) (*Hsiao et al., 2011*; *Korada et al., 2013*; *Hsiao et al., 2012*). RNase T and ExoI accommodate longer RNA substrates, facilitated by an expansive active site. The structural analysis correlates closely with sequence conservation of the

phosphate cap motif, which is strict in Orn homologs but divergent in RNase T proteins (*Figure 2D* and *Figure 2—source data 2*). Our structural analysis also suggests that modifications at the bases may be tolerated consistent with the observation that all native bases can be accommodated at the active site (*Figure 2—figure supplement 2*). Modifications at the sugar or termini beyond a terminal 5'-phosphate will likely fit sub-optimally if at all.

Any attempts to obtain structures of Orn$_{Vc}$ with longer RNA substrates (3–5 bases in length) only resolved diribonucleotides at the active site (data not shown), further suggesting a narrow active site that is selective for dinucleotide species. These results are consistent with two recent reports of structural data for *Colwellia psychrerythraea* Orn and REXO2 bound to substrates (*Lee et al., 2019*; *Chu et al., 2019*). In the case of Orn$_{Cp}$, crystallization was attempted with a 5-mer RNA; however, only two bases were resolved at the active site (*Lee et al., 2019*). A similar observation was made with REXO2 for RNA substrates, but a complex with a longer DNA-based ligand was obtained (*Chu et al., 2019*), which may not allow conclusions for RNA substrates. In the same report, RNase activity against longer RNA was observed with REXO2, but required a 20x molar excess of enzyme over RNA, likely not representing physiological conditions (*Chu et al., 2019*).

## The phosphate cap is required for diribonuclease activity

To assess the impact on catalysis of the phosphate cap in comparison to other active site residues identified in the structural analysis, we introduced specific single-point mutations into Orn$_{Vc}$ (*Figure 3*). All protein variants expressed stably in *P. aeruginosa* (*Figure 3—figure supplement 2*). For two representative point mutants with alanine substitutions at a phosphate-cap or active-site/DEDD-motif residue, Orn$_{Vc}$-R$^{130}$A or Orn$_{Vc}$-D$^{12}$A, respectively, we showed that the purified proteins form stable dimers in solution, comparable to wild-type Orn$_{Vc}$ (*Figure 3—figure supplement 3*). Purified protein variants were evaluated for pGpG degradation. Mutations of the central phosphate cap residue R$^{130}$ to alanine led to complete loss of catalytic activity, comparable to mutants in the DEDD active site motif (*Figure 3A* and *Figure 3—figure supplement 4*) (*Orr et al., 2015*). These protein variants were similarly unable to cleave the 7-mer (*Figure 1—figure supplement 1*). We next asked whether Orn$_{Vc}$ with specific point mutations could complement the deletion of *orn* in *P. aeruginosa*. *P. aeruginosa* Δ*orn* accumulates pGpG that in turn inhibits c-di-GMP-specific phosphodiesterases (*Cohen et al., 2015*; *Orr et al., 2015*). As a net result, c-di-GMP accumulates in these cells, an effect that is associated with a hyper-biofilm and cell aggregation phenotype. While expression of wild-type Orn$_{Vc}$ complements the *P. aeruginosa* Δ*orn* resulting in a dispersed culture, complementation with variants that carry mutations in either the active site or phosphate cap residues results in cell aggregation indistinguishable from the Δ*orn* phenotype (*Figure 3B*). Together, these experiments demonstrate that an intact phosphate cap is required for enzyme function.

## Interaction of Orn with substrates

Structures of Orn$_{Vc}$ (and REXO2) with different diribonucleotides, including di-purine, di-pyrimidine and mixed substrates, revealed identical binding poses (*Figure 2—figure supplement 2*). Additional hydrogen bonding between purine residues and Orn in the 3' or 5' position correlates with a small, but detectable preference of Orn$_{Vc}$ for purine-containing substrates (*Figure 4*). Together, the structural analysis uncovered Orn's mode of substrate binding, which is conserved from bacteria to humans and indicates a unique selection for linear diribonucleotides with a 5' phosphate. The requirement for the 5' phosphate was tested with GpG as a competitor for Orn binding and cleavage of pGpG. Even in excess GpG, Orn bound pGpG and cleaved pGpG similar to untreated control (*Figure 4—figure supplement 1A B*) supporting the importance of the 5' phosphate cap.

Another distinguishing factor between Orn and other exoribonucleases with similar activities is the apparent inhibition of Orn by 3'-phosphoadenosine 5'-phosphate (pAp), a metabolic byproduct of sulfate assimilation that accumulates upon lithium poisoning of pAp-phosphatase (*Mechold et al., 2007*). DHH/DHHA-type oligoribonucleases such as NrnA from *Bacillus subtilis*, *Mycobacterium tuberculosis* and *Mycoplasma pneumoniae* are capable of dephosphorylating the pAp mononucleotide and their activity against oligoribonucleotides is unaffected by pAp (*Mechold et al., 2007*; *Postic et al., 2012*). In contrast, pAp has been described as a competitive inhibitor for *E. coli* Orn and the human REXO2 (*Mechold et al., 2006*). However, the oligo-cytosine RNAs used in these prior studies were labeled at their 5' end with a bulky fluorescent dye moiety (usually a cyanine or

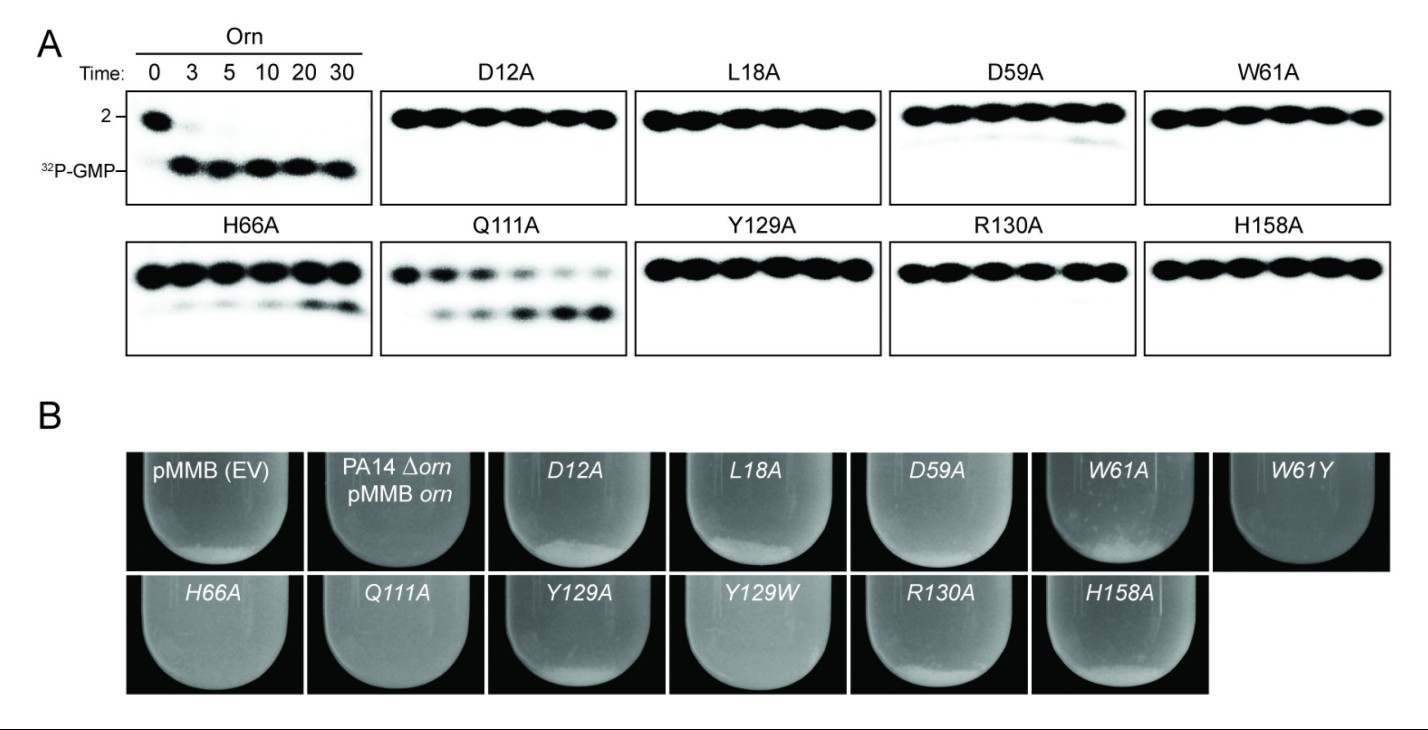

**Figure 3.** Functional importance of active-site and phosphate-cap residues for Orn function. (A) In vitro enzyme activity. Degradation of $^{32}$P-pGpG (1 µM) by purified wild-type Orn$_{Vc}$ or variants with alanine substitutions (5 nM) at the indicated sites was assessed. Samples were stopped at the indicated times (min) and analyzed by denaturing 20% PAGE. Representative gel images are shown with the indicated RNA size. *Figure 3—figure supplement 4* shows the graphs of the means and SD of three independent experiments. (B) In vivo activity of alanine substituted orn$_{Vc}$ alleles to complement *P. aeruginosa* Δorn. Overnight cultures of the indicated strains were allowed to stand for 10 min without agitation to allow bacterial aggregates to sediment. Representative images of the cultures of triplicated assays are shown.

The online version of this article includes the following source data and figure supplement(s) for figure 3:

**Figure supplement 1.** Contribution of catalytic residues and His$_{10}$-MBP tag to binding interaction between Orn$_{Vc}$ and pGpG.
**Figure supplement 2.** Immunoblot analysis of Orn in *P. aeruginosa* Δorn expressing indicated *orn* alleles.
**Figure supplement 3.** Molecular weight determination indicates that Orn$_{Vc}$ variants with point mutations at the phosphate cap and active site remain dimeric.
**Figure supplement 4.** pGpG degradation by purified wild-type Orn$_{Vc}$ or variants with indicated alanine substitutions.
**Figure supplement 4—source data 1.** Numerical data for *Figure 3—figure supplement 4*.

Cy-dye) (*Cohen et al., 2015*; *Mechold et al., 2006*); our aggregate data now indicate that these RNAs represent suboptimal substrates for Orn, given its stark requirement for a simple 5' phosphate (*Figure 2*). This labeling strategy is therefore likely to incompletely assess native Orn activity by underestimating diribonuclease activity while also overestimating the effect of competitive inhibitors. Revisiting pAp binding to Orn, we show here that Orn$_{Vc}$ does not interact with radiolabeled pAp in the binding assay that was used to quantify oligoribonucleotide interactions with Orn (*Figure 4—figure supplement 1C*). Furthermore, unlabeled pAp failed to competitively inhibit degradation of radiolabeled pGpG to GMP (*Figure 4—figure supplement 1A*).

## Orn is the only diribonuclease in *P. aeruginosa* grown under laboratory conditions

All available literature assumes Orn is a 3'−5' exoribonuclease that is responsible for processing of short (between 2 and 7) oligoribonucleotides. Yet our structural and biochemical data suggest that the enzyme exhibits such a striking preference for diribonucleotide substrates that the in vivo function of Orn as a general exoribonuclease should be reconsidered. Therefore, we developed experimental conditions to measure Orn's activity in cellular extracts. The *orn* gene is required for viability in most γ-proteobacteria, including *V. cholerae*; however, it is not essential for growth of *P.*

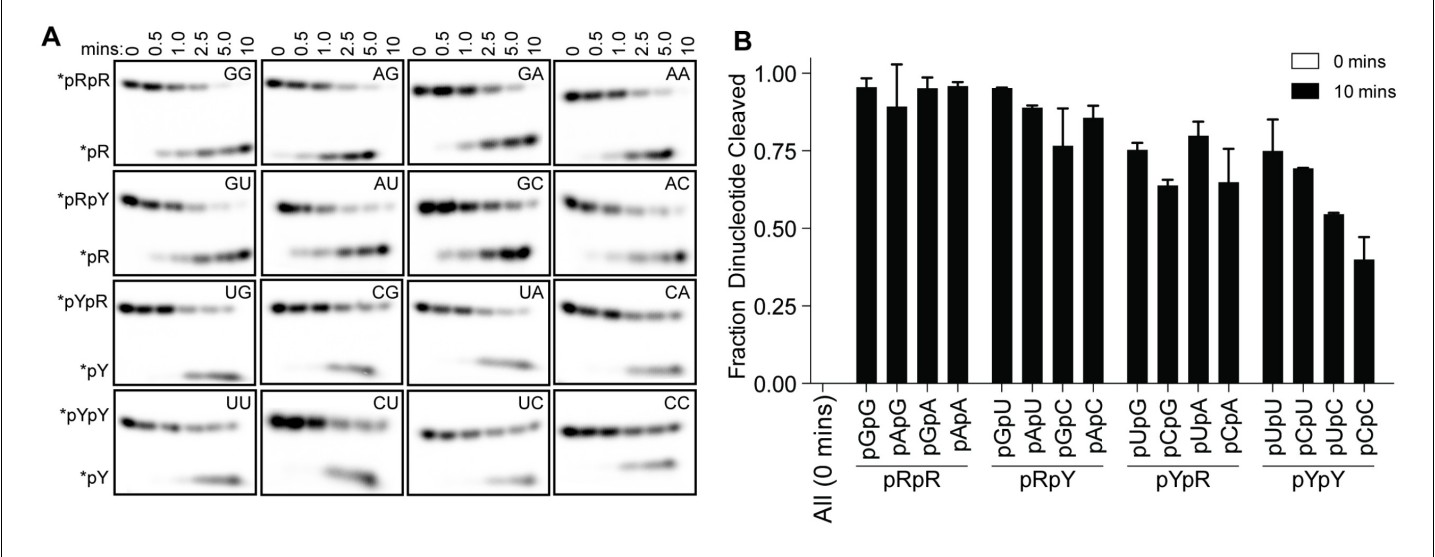

**Figure 4.** Orn$_{Vc}$ cleaves all diribonucleotides. (**A**) Orn$_{Vc}$ (5 nM) was incubated with di-purine (pRpR), purine-pyrimidine (pRpY or pYpR), or di-pyrimidine (pYpY) diribonucleotides (1 μM) containing the corresponding $^{32}$P-labeled RNA tracer. Aliquots of each reaction were stopped at the indicated times (min) and assessed by denaturing 20% PAGE. (**B**) Quantification of the intensities of bands corresponding to the amount of diribonucleotide cleaved at the 10 min time point. Results are the average and SD of duplicate independent experiments. Orn cleaves all diribonucleotides to nucleoside monosphosphates, albeit to varying extents. Diribonucleotides consisting of two purines (pRpR) were hydrolyzed most efficiently, with over 90% of starting RNAs processed by 10 min. A majority (>75%) of diribonucleotides with a 5′ purine (pRpY) were also processed by the 10 min endpoint. However, diribonucleotides with a 5′ pyrimidines exhibited moderately reduced levels of cleavage. Di-pyrimidine (pYpY) substrates, and in particular pUpC and pCpC, showed the slowest turnover from the substrates tested. These results demonstrate that while all diribonucleotides are acceptable substrates for Orn, the enzyme is likely to exhibit moderate preferences for diribonucleotides that contain a 5′ purine.

The online version of this article includes the following source data and figure supplement(s) for figure 4:

**Source data 1.** Numerical data for *Figure 4B*.
**Figure supplement 1.** Effect of GpG and pAp on Orn activity.
**Figure supplement 1—source data 1.** Numerical data for *Figure 4—figure supplement 1*.

*aeruginosa* under most conditions (*Cohen et al., 2015*; *Orr et al., 2015*). Therefore, lysates were generated from *P. aeruginosa* strains, including parental PA14, Δ*orn*, and Δ*orn* complemented with *orn$_{Vc}$*. 5′-$^{32}$P-radiolabeled 2-mer or 7-mer RNA was then added to each of these lysates (*Figure 5*). Aliquots of the mixtures were removed and analyzed by urea denaturing 20% PAGE at varying time intervals. Extracts from parental PA14 digested the entire radiolabeled diribonucleotide in less than 5 min (*Figure 5A*). In contrast, lysates from strains lacking Orn failed to show any signs of diribonuclease activity even at longer time points. Similarly, complementation with catalytically inactive alleles of *orn* (D$^{12}$A, L$^{18}$A and H$^{158}$A) also failed to clear the diribonucleotide (*Figure 5* and *Figure 5—figure supplement 1*). Diribonuclease activity in the lysates could be restored by ectopic expression of *orn* from a self-replicating plasmid or by addition of purified Orn (*Figure 5A*). These data confirm that cellular Orn is required for degrading diribonucleotides and no other cellular RNase of *P. aeruginosa* can substitute for Orn activity under these laboratory growth conditions.

When the $^{32}$P-7-mer RNA substrate was incubated in extracts from parental PA14 it was digested to a ladder of degradative intermediates including 6-mer, 5-mer, 4-mer and 3-mer RNAs (*Figure 5B*). Of note, while these intermediates and the final mononucleotide product accumulated over time, the 2-mer intermediate was never observed over the time course. In contrast, lysates from the Δ*orn* mutant specifically accrued the diribonucleotide intermediate with no apparent production of its mononucleotide products. Ectopic expression of plasmid-borne *orn*, but not *orn D$^{12}$A*, restored the degradation of the 7-mer to mononucleotides and the diribonucleotide intermediate could no longer be observed. Furthermore, the diribonucleotide intermediate that accumulated in the Δ*orn* lysate was fully processed upon addition of purified Orn protein (*Figure 5B*). Together, these results show that *P. aeruginosa* accumulates a bottleneck of diribonucleotide intermediates in a Δ*orn* background, which is only resolved upon addition of Orn. Degradation of RNA fragments

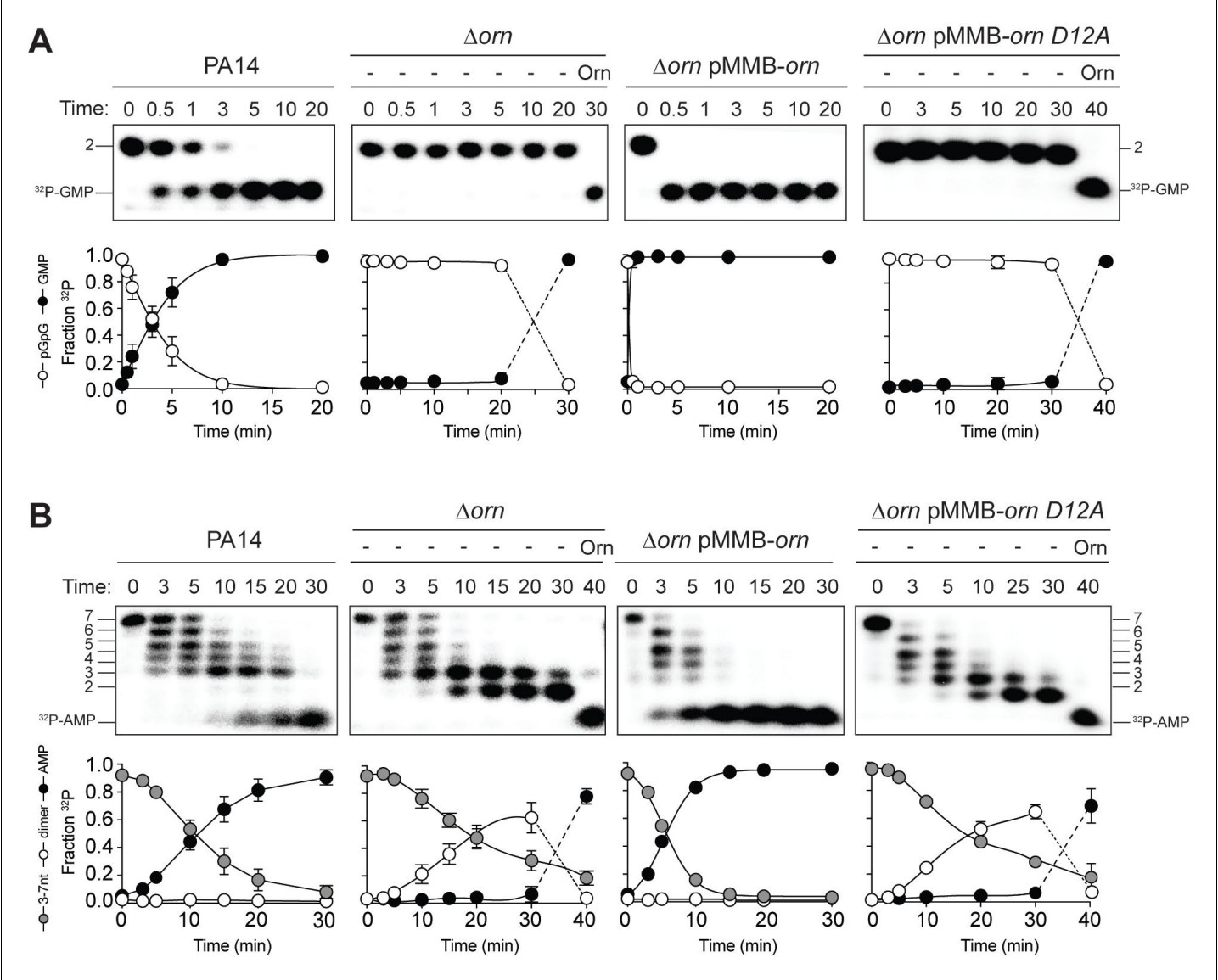

**Figure 5.** Orn acts as a diribonuclease in cell lysates. Degradation of [32]P-GG (**A**) and [32]P-AAAAAGG (**B**) by whole cell lysates of wild-type, *orn* mutant, *orn* mutant complemented with *orn*$_{Vc}$, or *orn*$_{Vc}$ $D^{12}A$. For Δ*orn* and Δ*orn* complemented with *orn*$_{Vc}$ $D^{12}A$,100 nM of purified Orn$_{Vc}$ was added at 30 min time point and incubated for an additional 10 min. Samples were stopped at the indicated time and analyzed by 20% denaturing PAGE. Representative gel images of triplicated assays are shown with the indicated RNA size. Graphs show quantitation of triplicate data for indicated RNA species over time. The online version of this article includes the following source data and figure supplement(s) for figure 5:

**Source data 1.** Numerical data for *Figure 5*.
**Figure supplement 1.** Catalytic residues of Orn$_{Vc}$ are required for degradation for [32]P-AAAAAGG in whole-cell lysates.

with three or more residues by Orn is negligible in a cellular context, considering that Δ*orn* lysates preserve nuclease activities for the processing of RNAs down to diribonucleotides. From these aggregate data, we propose that Orn functions not as an oligoribonuclease as stated in the literature but instead functions as a specialized ribonuclease of diribonucleotide substrates (i.e. a 'diribonuclease').

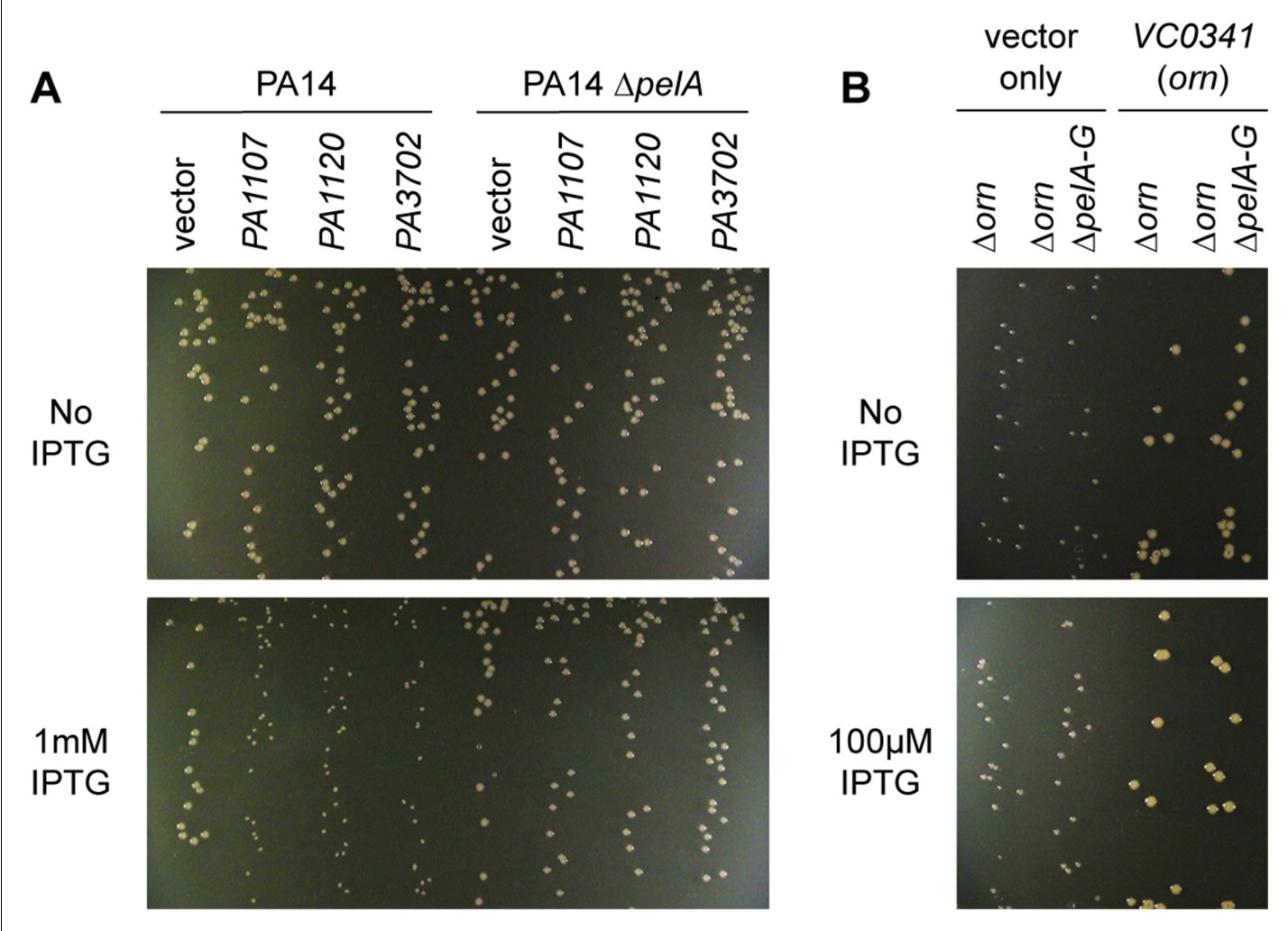

**Figure 6.** Small-colony phenotype of Δ*orn* is independent of c-di-GMP signaling. Bacterial cultures were diluted and dripped on LB agar plates with the indicated concentration of IPTG. After overnight incubation, representative images of the plates were taken and shown for (**A**) PA14 and PA14 Δ*pelA* harboring pMMB vector with the indicated diguanylate cyclase gene and (**B**) PA14 Δ*orn* and PA14 Δ*orn* Δ*pelA-G* with pMMB and pMMB-*orn*. Experiments were performed in triplicate.

The online version of this article includes the following figure supplement(s) for figure 6:

**Figure supplement 1.** Small-colony phenotype of Δ*orn* is complemented by functional *orn*$_{Vc}$ alleles.

**Figure supplement 2.** *P. aeruginosa* PAK Δ*orn* mutants show a small-colony phenotype.

## The diribonucleotide pool affects *P. aeruginosa* growth in addition to c-di-GMP signaling

We noticed that PA14 Δ*orn* had reduced growth on agar plates that is reminiscent of a small colony variant (SCV) phenotype (*Figure 6*) (*D'Argenio et al., 2002*). The SCV phenotype has been attributed previously to increased c-di-GMP levels (*Hickman et al., 2005*). C-di-GMP binds to FleQ and activates transcription of the *pel* operon (*Baraquet et al., 2012*; *Matsuyama et al., 2016*). In addition, c-di-GMP binds to the c-di-GMP-receptor PelD to increase the biosynthesis of the pel exopolysaccharide (*Lee et al., 2007*), which enhances cell aggregation leading to a compact SCV morphology. Previous reports had shown that *P. aeruginosa* Δ*orn* is unable to clear pGpG, which results in the elevation of c-di-GMP signaling and pel-dependent cell aggregation and biofilm mass (*Cohen et al., 2015*; *Orr et al., 2015*).

To determine whether the functional impact of diribonucleotide build-up is due specifically to an increased pGpG level and its effect on c-di-GMP, we asked whether the SCV formation in the Δ*orn*

mutant is dependent on c-di-GMP processes. First, we confirmed that SCV formation in PA14 can be induced by elevated c-di-GMP levels through overexpression of diguanylate cyclases, such as PA1107 (*roeA*), PA1120 (*yfiN/tpbB*) and PA3702 (*wspR*) (*Figure 6A*) (*Hickman et al., 2005*; *Kulasakara et al., 2006*; *Ueda and Wood, 2009*; *Malone et al., 2010*). As expected, the effect of c-di-GMP is due to increased production of the pel exopolysaccharide and biofilm formation since the PA14 *ΔpelA* mutant maintained normal colony morphology even when these diguanylate cyclases were overexpressed (*Figure 6A*). To determine whether the increase in the pool of pGpG and c-di-GMP is the only reason for small-colony growth in the *Δorn* mutant, a *Δorn Δpel* double mutant was tested. When *Δorn Δpel* was grown on agar, it also had an apparent SCV phenotype (*Figure 6B*). Complementation of *Δorn* and *Δorn Δpel* with active alleles of *orn* restored normal colony morphology, whereas inactive *orn* alleles failed to complement (*Figure 6—figure supplement 1*). In every case, the colony morphology was the same between *Δorn* and *Δorn Δpel* indicating a second pathway that can restrict colony growth, but in this case independent of *pel* and c-di-GMP. Similar results were obtained for *P. aeruginosa* PAK strain (*Figure 6—figure supplement 2*). These results indicate that increased pools of one or more of the diribonucleotides function to cause small-colony growth in *Δorn* in addition to the actions of pGpG on c-di-GMP signaling.

## Conclusion

Orn is unique amongst exoribonucleases because it is essential in some γ-proteobacteria and is required to degrade the pGpG intermediate in c-di-GMP signaling. Yet the molecular and structural requirements for Orn were unknown. Our studies reveal that Orn is a dedicated diribonuclease in cells. This appears to be driven by a catalytic site that is restricted by a cap that mediates multiple interactions with the 5' phosphate of diribonucleotide substrates. This constriction prevents longer substrates from binding with high affinity, rendering them poor substrates for catalytic cleavage. The discovery that Orn acts as a dedicated diribonuclease indicates that this activity clears a specific diribonucleotide bottleneck in global RNA degradation (*Figure 7*). Prior studies have shown that *orn*

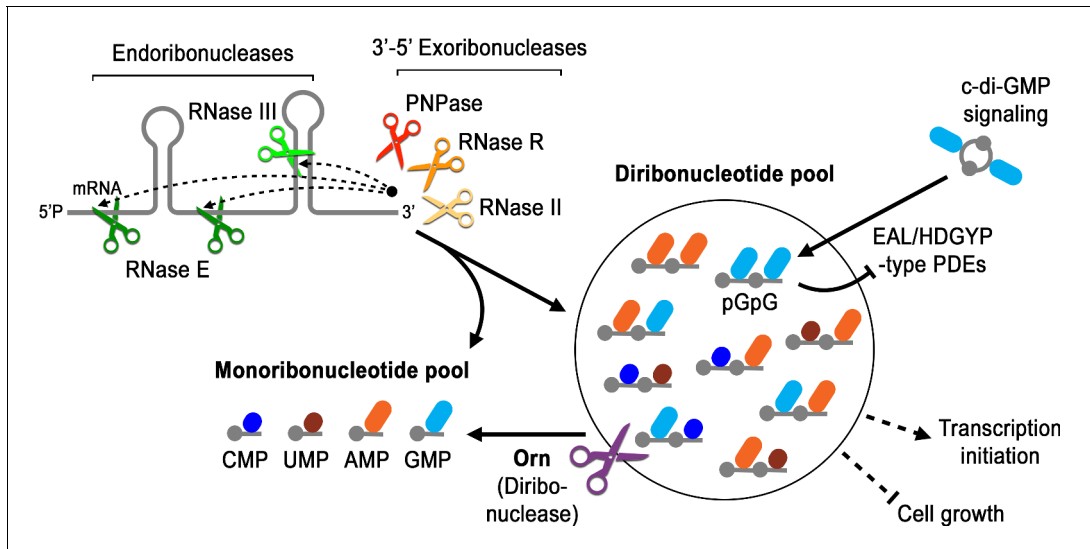

**Figure 7.** Model for Orn's cellular function as a diribonuclease. RNA degradation is initiated by fragmentation via endoribonucleases (e.g. RNase E and RNase III) that cleave unstructured or structured RNA sequences. RNA fragments are processed further at their 3' termini by 3'−5' exoribonucleases (e.g. PNPase, RNase R, and RNase II). Their combined activity produces mononucleotides and various terminal diribonucleotides from the original RNA fragments. The pGpG (GG) linear diribonucleotide is also produced by linearization of c-di-GMP by specific phosphodiesterases, EAL- and HD-GYP-domain-containing enzymes, which terminate c-di-GMP signaling. In *Pseudomonas aeruginosa* and likely other organisms that rely on Orn for growth, Orn is the only diribonuclease that cleaves diribonucleotides to mononucleotides. In the absence of Orn, diribonucleotides accumulate with a drastic impact on cellular physiology, ranging from transcriptional changes, small-colony phenotypes and growth arrest, depending on the organism. Orn is also unique because it acts as the second phosphodiesterase in the degradation of c-di-GMP by clearing the pGpG intermediate. In an *orn* mutant, c-di-GMP levels are elevated through feedback inhibition of the c-di-GMP-degrading phosphodiesterases by pGpG, leading to the associated biofilm phenotypes.

depletion can lead to accumulation of diribonucleotides and longer oligoribonucleotides in some cellular backgrounds (*Ghosh and Deutscher, 1999*). The increase in oligoribonucleotides longer than dimers in cells may occur through feedback inhibition of other enzymes in the RNA degradation pathway, in analogy to the impact of pGpG on c-di-GMP-degrading phosphodiesterases and c-di-GMP signaling (*Figure 7*) (*Cohen et al., 2015*; *Orr et al., 2015*). Whether the effect of diribonucleotides on essentiality is due to general processes such as RNA degradation, altered transcription (*Goldman et al., 2011*; *Druzhinin et al., 2015*; *Vvedenskaya et al., 2012*), specific interactions with essential proteins, or a combination of these remains to be evaluated in this context. Our studies therefore reveal a key step in RNA degradation, the enzymatic cleavage of diribonucleotides into mononucleotides, and set the stage to address how diribonucleotide accumulation is detrimental to cell survival.

# Materials and methods

## Key resources table

| Reagent type or resource | Designation | Source or reference | Identifiers | Additional information |
|---|---|---|---|---|
| *Strain* (*P. aeruginosa*) | PA14 | *Rahme et al., 1995*; PMID 7604262 | | |
| *Strain* (*P. aeruginosa*) | PA14 ΔpelA | *Lee et al., 2007*; PMID: 17824927 | | |
| *Strain* (*P. aeruginosa*) | PA14 Δorn | *Orr et al., 2015*; PMID: 26305945 | | |
| *Strain* (*P. aeruginosa*) | PA14 Δorn ΔpelA-G | *Orr et al., 2015*; PMID: 26305945 | | |
| *Strain* (*P. aeruginosa*) | PAK | *Bradley, 1974*; PMID: 4206974 | | |
| *Strain* (*P. aeruginosa*) | PAK Δorn | This study | | Generated using pEX-Gn-Δorn (PAK) |
| Strain (*E. coli*) | XL10-Gold | Agilent | | |
| Strain (*E. coli*) | Stellar cells | Takara/Clontech | | |
| Strain (*E. coli*) | BL21(DE3) | New England Biolabs | | |
| Strain (*E. coli*) | NEB T7Iq | New England Biolabs | | |
| Genetic reagent (plasmid) | pET28-His6-SUMO-OrnVc | This study | | cloned from custom DNA fragment (see below) for purification of His6-SUMO-OrnVc, OrnVc |
| Genetic reagent (plasmid) | pET28-His6-SUMO-Rexo2 | This study | | cloned from custom DNA fragment (see below) for purification of His6-SUMO-Rexo2, Rexo2 |
| Genetic reagent (plasmid) | pDONR221-VC0341 (ornVc) | *Rolfs et al., 2008*; PMID: 18337508 | | |
| Genetic reagent (plasmid) | pDONR221-ornVc D12A | This study | | site-directed mutagenesis to introduce alanine at D12 position |
| Genetic reagent (plasmid) | pDONR221-ornVc L18A | This study | | site-directed mutagenesis to introduce alanine at L18 position |
| Genetic reagent (plasmid) | pDONR221-ornVc D59A | This study | | site-directed mutagenesis to introduce alanine at D59 position |

*Continued on next page*

*Continued*

| Reagent type or resource | Designation | Source or reference | Identifiers | Additional information |
|---|---|---|---|---|
| Genetic reagent (plasmid) | pDONR221-ornVc Q111A | This study | | site-directed mutagenesis to introduce alanine at Q111 position |
| Genetic reagent (plasmid) | pDONR221-ornVc R130A | This study | | site-directed mutagenesis to introduce alanine at R130 position |
| Genetic reagent (plasmid) | pDONR221-ornVc Y129A | This study | | site-directed mutagenesis to introduce alanine at Y129 position |
| Genetic reagent (plasmid) | pDONR221-ornVc Y129W | This study | | site-directed mutagenesis to introduce tryptophan at Y129 position |
| Genetic reagent (plasmid) | pDONR221-ornVc W61A | This study | | site-directed mutagenesis to introduce alanine at W61 position |
| Genetic reagent (plasmid) | pDONR221-ornVc W61Y | This study | | site-directed mutagenesis to introduce tyrosine at W61 position |
| Genetic reagent (plasmid) | pDONR221-ornVc H66A | This study | | site-directed mutagenesis to introduce alanine at H66 position |
| Genetic reagent (plasmid) | pDONR221-ornVc H158A | This study | | site-directed mutagenesis to introduce alanine at H158 position |
| Genetic reagent (plasmid) | pVL847-ornVc | *Orr et al., 2015*; PMID: 26305945 | | OrnVc from pDONR cloned into pVL847 for purification of His-MBP-OrnVc, OnrVc |
| Genetic reagent (plasmid) | pVL847-ornVc D12A | This study | | D12A from pDONR cloned into pVL847 for purification of His-MBP-OrnVc D12A, OrnVc D12A |
| Genetic reagent (plasmid) | pVL847-ornVc L18A | This study | | L18A from pDONR cloned into pVL847 for purification of His-MBP-OrnVc L18A, OrnVc L18A |
| Genetic reagent (plasmid) | pVL847-ornVc D59A | This study | | L18A from pDONR cloned into pVL847 for purification of His-MBP-OrnVc D59A, OrnVc D59A |
| Genetic reagent (plasmid) | pVL847-ornVc Q111A | This study | | Q111A from pDONR cloned into pVL847 for purification of His-MBP-OrnVc Q111A, OrnVc Q111A |
| Genetic reagent (plasmid) | pVL847-ornVc R130A | This study | | R130A from pDONR cloned into pVL847 for purification of His-MBP-OrnVc R130A, OrnVc R130A |
| Genetic reagent (plasmid) | pVL847-ornVc Y129A | This study | | Y129A from pDONR cloned into pVL847 for purification of His-MBP-OrnVc Y129A, OrnVc Y129A |
| Genetic reagent (plasmid) | pVL847-ornVc W61A | This study | | W61A from pDONR cloned into pVL847 for purification of His-MBP-OrnVc W61A, OrnVc W61A |
| Genetic reagent (plasmid) | pVL847-ornVc H66A | This study | | H66A from pDONR cloned into pVL847 for purification of His-MBP-OrnVc H66A, OrnVc H66A |
| Genetic reagent (plasmid) | pVL847-ornVc H158A | This study | | H158A from pDONR cloned into pVL847 for purification of His-MBP-OrnVc H158A, OrnVc H158A |
| Genetic reagent (plasmid) | pMMB-Gn | *Fürste et al., 1986*; PMID: 3549457 | | |

*Continued on next page*

*Continued*

| Reagent type or resource | Designation | Source or reference | Identifiers | Additional information |
|---|---|---|---|---|
| Genetic reagent (plasmid) | pMMB-Gn-PA1107 | *Kulasakara et al., 2006*; PMID: 16477007 | | |
| Genetic reagent (plasmid) | pMMB-Gn-PA1120 | *Kulasakara et al., 2006*; PMID: 16477007 | | |
| Genetic reagent (plasmid) | pMMB-Gn-PA3702 (wspR) | *Kulasakara et al., 2006*; PMID: 16477007 | | |
| Genetic reagent (plasmid) | pMMB-Ap | *Fürste et al., 1986*; PMID: 3549457 | | |
| Genetic reagent (plasmid) | pMMB-Ap-VC0341 (ornVc) | This study | | Orn from pDONR cloned into pMMB for expressing in *P. aeruginosa* |
| Genetic reagent (plasmid) | pMMB-Ap-ornVc D12A | This study | | D12A from pDONR cloned into pMMB for expressing in *P. aeruginosa* |
| Genetic reagent (plasmid) | pMMB-Ap-ornVc L18A | This study | | L18A from pDONR cloned into pMMB for expressing in *P. aeruginosa* |
| Genetic reagent (plasmid) | pMMB-Ap-ornVc Q111A | This study | | Q111A from pDONR cloned into pMMB for expressing in *P. aeruginosa* |
| Genetic reagent (plasmid) | pMMB-Ap-ornVc R130A | This study | | R130A from pDONR cloned into pMMB for expressing in *P. aeruginosa* |
| Genetic reagent (plasmid) | pMMB-Ap-ornVc Y129A | This study | | Y129A from pDONR cloned into pMMB for expressing in *P. aeruginosa* |
| Genetic reagent (plasmid) | pMMB-Ap-ornVc Y129W | This study | | Y129W from pDONR cloned into pMMB for expressing in *P. aeruginosa* |
| Genetic reagent (plasmid) | pMMB-Ap-ornVc W61A | This study | | W61A from pDONR cloned into pMMB for expressing in *P. aeruginosa* |
| Genetic reagent (plasmid) | pMMB-Ap-ornVc W61Y | This study | | W61Y from pDONR cloned into pMMB for expressing in *P. aeruginosa* |
| Genetic reagent (plasmid) | pMMB-Ap-ornVc H66A | This study | | H66A from pDONR cloned into pMMB for expressing in *P. aeruginosa* |
| Genetic reagent (plasmid) | pMMB-Ap-ornVc H158A | This study | | H158A from pDONR cloned into pMMB for expressing in *P. aeruginosa* |
| Genetic reagent (plasmid) | pEX-Gn-Δorn (PAK) | This study | | |
| Genetic reagent (plasmid) | pJHA-Gn | This study | | |
| Genetic reagent (plasmid) | pJHA-Gn-ornPa | This study | | OrnPA (Orr, et al, PNAS 2015) cloned into pJHA for expressing in *P. aeruginosa* |
| Genetic reagent (plasmid) | pJHA-Gn-ornVc | This study | | Orn from pDONR cloned into pJHA for expressing in *P. aeruginosa* |
| Genetic reagent (plasmid) | pJHA-Gn-ornVc D12A | This study | | D12A from pDONR cloned into pJHA for expressing in *P. aeruginosa* |
| Genetic reagent (plasmid) | pJHA-Gn-ornVc L18A | This study | | L18A from pDONR cloned into pJHA for expressing in *P. aeruginosa* |
| Genetic reagent (plasmid) | pJHA-Gn-ornVc D59A | This study | | Q111A from pDONR cloned into pJHA for expressing in *P. aeruginosa* |
| Genetic reagent (plasmid) | pJHA-Gn-ornVc W61A | This study | | R130A from pDONR cloned into pJHA for expressing in *P. aeruginosa* |
| Genetic reagent (plasmid) | pJHA-Gn-ornVc H66A | This study | | Y129A from pDONR cloned into pJHA for expressing in *P. aeruginosa* |
| Genetic reagent (plasmid) | pJHA-Gn-ornVc Q111A | This study | | W61A from pDONR cloned into pJHA for expressing in *P. aeruginosa* |

*Continued on next page*

*Continued*

| Reagent type or resource | Designation | Source or reference | Identifiers | Additional information |
|---|---|---|---|---|
| Genetic reagent (plasmid) | pJHA-Gn-ornVc R130A | This study | | H66A from pDONR cloned into pJHA for expressing in *P. aeruginosa* |
| Genetic reagent (plasmid) | pJHA-Gn-ornVc H158A | This study | | H158A from pDONR cloned into pJHA for expressing in *P. aeruginosa* |
| Antibody | Mouse anti-HA monoclonal antibody | Sigma; catalog number H9658 | | 1:5000 |
| Antibody | Goat Anti-Mouse IgG-HRP conjugate | Sigma; catalog number A9917 | | 1:5000 |
| DNA fragment | 5'-GGATCCATGGCAGCCGGTGAAAG CATGGCACAGCGTATGGTTTGGGTT GATCTGGAAATGACCGGTCTGGATA TTGAAAAAGATCAGATTATTGAAATG GCCTGCCTGATTACCGATAGCGATCT GAATATTCTGGCAGAAGGTCCGAAT CTGATTATCAAACAGCCGGATGAACT GCTGGATAGCATGAGCGATTGGTGT AAAGAACATCATGGTAAAAGCGGTCT GACCAAAGCAGTTAAAGAAAGCACCA TTACACTGCAGCAGGCCGAATATGAAT TTCTGAGCTTTGTTCGTCAGCAGACC CCTCCGGGTCTGTGTCCGCTGGCAGG TAATAGCGTTCATGAAGATAAAAAGTT TCTGGATAAGTATATGCCGCAGTTTAT GAAGCATCTGCATTATCGCATTATTGAT GTGAGCACCGTTAAAGAACTGTGTCGT CGTTGGTATCCGGAAGAATATGAGTTT GCACCGAAAAAAGCAGCAAGCCATCGT GCACTGGATGATATTAGCGAAAGCATC AAAGAGCTGCAGTTTTATCGCAACAAC ATCTTCAAAAAGAAAATCGACGAGAAAA AACGCAAAATCATCGAAAACGGCGAAAA CGAAAAAACCGTTAGCTAAGCGGCCGC-3' | GeneArt | | custom DNA fragment for REXO2 that is codon-optimized for *E. coli* |
| DNA fragment | 5'-GGATCCATGAGCTTTAGCGATCAGAAT CTGATTTGGATTGATCTGGAAATGGACCG GTCTGGACCCGGAAATGCATAAAATCAT TGAAATGGCAACCATCGTGACCGATAGCG AACTGAATATTCTGGCAGAAGGTCCGGTT ATTGCAATTCATCAGCCGGAAAGCGAACT GGCAAAAATGGATGAATGGTGTACCACCA CCCATACCGCAAGCGGTCTGGTTGCACGT GTTCGTCAGAGCCAGGTTAGCGAAGAAGA AGCAATTGATCAGACCCTGGCATTTCTGAA ACAGTGGGTTCCGGAAGGTAAAAGCCCGAT TTGTGGTAATAGCATTGGTCAGGATCGTCG CTTTCTGTATAAACATATGCCTCGTCTGGAA GCCTATTTCCATTATCGTTATATTGATGTGAG CACCATCAAAGAACTGACCCGTCGTTGGCAG CCGGAAGTTCTGAAAGAATTTAGCAAAACCG GTAGCCATCTGGCACTGGATGATATTCGTGA AAGCATTGCAGAGCTGCAGTTTTATCGTAAA GCCGTGTTTAAAATCTAAGCGGCCGC-3' | GeneArt | | custom DNA fragment for OrnVc that is codon-optimized for *E. coli* |
| RNA primer | 5'-GG-3' | Sigma | | |
| RNA primer | 5'-AGG-3' | Sigma | | |
| RNA primer | 5'-AAGG-3' | Sigma | | |
| RNA primer | 5'-AAAGG-3' | Sigma | | |
| RNA primer | 5'-AAAAGG-3' | Sigma | | |
| RNA primer | 5'-AAAAAGG-3' | Sigma | | |
| RNA primer | 5'-pGG-3' | Biolog; catalog number P023-01 | | |

*Continued on next page*

*Continued*

| Reagent type or resource | Designation | Source or reference | Identifiers | Additional information |
|---|---|---|---|---|
| RNA primer | 5'-pAA-3' | Biolog; catalog number P033-01 | | |
| RNA primer | 5'-pAG-3' | GE Healthcare Dharmacon | | |
| RNA primer | 5'-pGA-3' | GE Healthcare Dharmacon | | |
| RNA primer | 5'-pGC-3' | GE Healthcare Dharmacon | | |
| RNA primer | 5'-pCG-3' | GE Healthcare Dharmacon | | |
| RNA primer | 5'-pCU-3' | GE Healthcare Dharmacon | | |
| Software | Prism | GraphPad | | |
| Software | XDS | *Kabsch, 2010*; PMID: 20124693 | | Distributed through SBGrid |
| Software | Pointless | *Evans, 2006*; PMID: 16369096 | | Distributed through SBGrid |
| Software | Scala | *Evans, 2006*; PMID: 16369096 | | Distributed through SBGrid |
| Software | Phenix | *Adams et al., 2010*; PMID: 20124702 | | Distributed through SBGrid |
| Software | Coot | *Emsley et al., 2010*; PMID: 20383002 | | Distributed through SBGrid |
| Software | Pymol | Schrödinger | | |
| Software | Fujifilm Multi Gauge software v3.0 | Fujifilm | | |

## Cloning, protein expression and purification

*orn* genes from *Vibrio cholerae* O1 El Tor VC0341 (residues 1–181) and *Homo sapiens* REXO2 (residues 33–237) were synthesized by Geneart (Life Technologies). Genes were cloned by ligation between BamHI and NotI sites of a modified pET28a vector (Novagen) yielding N-terminally His$_6$-tagged small ubiquitin-like modifier (SUMO) fusion proteins cleavable by recombinant Ulp-1 protease.

Orn proteins were overexpressed in *E. coli* BL21 T7 Express cells (New England Biolabs). Fresh transformants were grown in Terrific Broth (TB) supplemented with 50 ug/mL kanamycin at 37°C to an OD$_{600}$ ~1.0, at which point the temperature was reduced to 18°C and expression was induced by addition of 0.5 mM IPTG. Cells were harvested after 16 hr of expression by centrifugation and resuspended in a minimal volume of Ni-NTA binding buffer (25 mM Tris-Cl, 500 mM NaCl, 20 mM imidazole, pH 8.5) followed by flash freezing in liquid nitrogen.

Cells were thawed and lysed by sonication. Cell debris was removed by centrifugation and clarified soluble lysate was incubated with Ni-NTA resin (Qiagen) pre-equilibrated with Ni-NTA binding buffer. Following one hour of binding, the resin was washed three times with 10 column volumes of Ni-NTA binding buffer, and then eluted with six column volumes of Ni-NTA elution buffer (25 mM Tris-Cl, 500 mM NaCl, 350 mM imidazole, pH 8.5). Eluates were buffer exchanged into gel filtration buffer (25 mM Tris-Cl, 150 mM NaCl, pH 7.5) via a HiPrep 26/10 desalting column (GE Healthcare), followed by overnight incubation with Ulp-1 to cleave off the His$_6$-tagged SUMO moiety. Untagged Orn proteins were recovered in the flow-through of a HisTrap Ni-NTA column (GE Healthcare), separated from His$_6$-SUMO, uncleaved proteins and His$_6$-tagged Ulp-1. Orn was concentrated via Amicon Ultra 10K concentrator prior to loading onto a HiLoad 16/60 Superdex 200 gel filtration column (GE Healthcare) equilibrated in gel filtration buffer. Fractions containing Orn were pooled and concentrated to 100 mg/mL, frozen in liquid nitrogen, and stored at −80°C.

## Bacterial strains, plasmids and growth conditions

Strains and plasmids are listed in Key Resources table. *P. aeruginosa* is routinely grown in LB supplemented with the appropriate antibiotic (50 µg/mL carbenicillin or 75 µg/mL gentamicin) at 37°C.

## Protein crystallography

REXO2-RNA and Orn$_{Vc}$-RNA complexes (pGpG and pApA from BioLog Life Science Institute, other nucleotides from GE Healthcare Dharmacon) were formed prior to crystallization by mixing a 1:2 molar ratio of Orn:RNA in gel filtration buffer, followed by incubation for 30 min at the crystallization temperature. Orn-RNA complexes (10–30 mg/ml) were crystallized via hanging-drop vapor diffusion by mixing equal volumes (0.8 µl) of sample with reservoir solution. Orn$_{Vc}$ crystals grew at 20°C over a reservoir solution that was composed of 0.1 M BisTris (pH 5.5), 17% polyethylene glycol 3350, and 20% xylitol. Crystals were flash-frozen in liquid nitrogen. REXO2 crystals grew at 4°C, using a reservoir comprised of 0.2 M sodium malonate (pH 5.5) and 15–20% polyethylene glycol 3350. REXO2 crystals were soaked in cryoprotectant of reservoir solution supplemented with 25% glycerol prior to flash freezing with liquid nitrogen. All crystals were stored in liquid nitrogen. Data were collected by synchrotron radiation at 0.977 Å on frozen crystals at 100 K at beamline F1 of the Cornell High Energy Synchrotron Source (CHESS). Diffraction data sets were processed using XDS, Pointless, and Scala (*Evans, 2006*; *Kabsch, 2010*). The initial structures were solved by Molecular Replacement using the software package Phenix (*Adams et al., 2010*) and the unpublished coordinates of *E. coli* Orn (PDB: 2igi) as the search model. Manual model building and refinement were carried out with Coot (*Emsley et al., 2010*) and Phenix, respectively. Illustrations were prepared in Pymol (Version 2.2.0, Schrodinger, LLC). All software packages were accessed through SBGrid (*Morin et al., 2013*). Crystallographic statistics are shown in *Figure 2—source data 1*.

## Site-directed mutagenesis

To create the point mutants of *orn$_{Vc}$*, mutations were generated by using the Q5 Site-Directed Mutagenesis Kit (New England Biolabs). All mutations were verified by sequencing.

## Western blot analysis

Overnight cultures were diluted to OD$_{600}$ of ~0.02 in media with 0.2% arabinose with 75 µg/mL gentamicin. Culture density was monitored by OD$_{600}$. At OD$_{600}$ ~1.5, cells were collected by centrifugation and resuspended in 1/15 vol of 1x PBS. Proteins were immediately precipitated by a modified MeOH/CHCl$_3$ procedure (sample/MeOH/CHCl$_3$: 1/1/0.25 [*Wessel and Flügge, 1984*]). Samples were separated on 12% SDS-PAGE, transferred to PVDF membranes and blocked with Tris buffered saline (50 mM Tris, pH7.4, 200 mM NaCl) with 5% non-fat milk. HA-tags were detected using mouse anti-HA (Sigma H9658) and goat anti-mouse IgG-HRP conjugate (Sigma A9917). HRP signal was developed using chemiluminescence per manufacture protocol (Sigma).

## Labeling of RNAs

5' un-phosphorylated RNAs were purchased from TriLink Biotechnologies or Sigma. Each RNA was subjected to radioactive end-labeling or non-radioactive phosphorylation by T4 Polynucleotide Kinase (New England Biolabs). Each RNA was subjected to phosphorylation with equimolar concentrations of either $^{32}$P-γ-ATP or ATP, T4 PNK, and 1X T4 PNK Reaction Buffer. Reactions comprising a final concentration of either 0.5 µM radiolabeled RNA or 2.0 µM phosphorylated RNA were incubated at 37°C for 40 min, followed by heat inactivation of T4 PNK at 65°C for 20 min.

## Protein expression and purification for biochemical assays

*E. coli* T7Iq strains harboring expression vector pVL847 expressing an His$_{10}$-MBP-Orn and His$_{10}$-MBP-Orn mutants from *V. cholerae* were grown overnight, subcultured in LB M9 fresh media supplemented with 15 µg/ml gentamicin and grown to approximately OD$_{600}$0.5 ~ 1.0 at 30°C. Expression was induced with 1 mM IPTG for 4 hr. Induced bacteria were collected by centrifugation and resuspended in 10 mM Tris, pH 8, 100 mM NaCl, and 25 mM imidazole. After addition of 10 µg/mL DNase, 25 µg/mL lysozyme, and 1 mM PMSF, bacteria were lysed by sonication. Insoluble material was removed by centrifugation. The His-fusion protein was purified by separation over a Ni-NTA column. Purified proteins were pooled and dialyzed for 1 hr and overnight against 10 mM Tris, pH 8,

100 mM NaCl. The proteins were dialyzed for 3 hr in 10 mM Tris, pH 8, 100 mM NaCl, and 50% (vol/vol) glycerol, aliquoted, and flash frozen with liquid nitrogen and stored at −80˚ ˚C.

## Size-exclusion chromatography coupled multiangle light scattering (SEC-MALS)

5 mg/ml (0.24 mM) of purified protein (wild-type $Orn_{Vc}$, $Orn_{Vc}$-$D^{12}A$, or $Orn_{Vc}$-$R^{130}A$) was injected onto a Superdex 200 Increase 10/300 column (GE Healthcare) equilibrated with gel filtration buffer (25 mM Tris-Cl, pH 7.5, 150 mM NaCl). Samples were run continuously at a flow rate of 0.75 ml/min through the gel filtration column coupled to a static 18-angle light scattering detector (DAWN HELIOS-II) and a refractive index detector (Optilab T-rEX), with data being collected every second. Data analysis was performed with ASTRA VI, yielding the molar mass and mass distribution (polydispersity) of the sample, using monomeric BSA (Sigma; 5 mg/ml) to normalize the light scattering detectors and as a control sample.

## Preparation of whole cell lysates

Overnight cultures of *P. aeruginosa* PA14 WT, Δ*orn* mutant, or complemented strains were subcultured into fresh media with antibiotic and 1 mM IPTG, grown at 37˚C with shaking to $OD_{600}$ ~ 2.5. All bacteria samples were collected by centrifuge and resuspended in 1/10 vol of 100 mM NaCl, and 100 mM Tris, pH 8, also supplemented with 25 µg/mL lysozyme, 10 µg/mL DNase, and 1 mM PMSF and stored at −80˚C.

## Oligoribonuclease cleavage reactions

Phosphorylated RNA (1.0 µM), including trace amounts of radiolabeled substrate, was subjected to cleavage by either 5.0 nM or 1.0 µM purified Orn at room temperature. These reactions were in the presence of 10 mM Tris, pH 8.0, 100 mM NaCl, and 5 mM $MgCl_2$. At the appropriate times, aliquots of the reaction were removed and quenched in the presence of 150 mM EDTA on ice and heat inactivated at 95˚C for 5 min. Activity of whole cell lysates against $^{32}P$-labeled oligoribonucleotide substrates was performed at room temperature in reaction buffer (10 mM Tris, pH 8, 100 mM NaCl, and 5 mM $MgCl_2$). At the indicated times, the reaction was stopped by the addition of 0.2 M EDTA and heated at 98˚C for 5 min. Samples were separated on denaturing 20% PAGE containing 1x TBE and 4 M urea. The gels were imaged using Fujifilm FLA-7000 phosphorimager (GE) and analyzed for the appearance of truncated $^{32}P$-labeled products. The intensity of the radiolabeled nucleotides was quantified using Fujifilm Multi Gauge software v3.0.

## DRaCALA measurement of dissociation constants

To measure $K_d$, serial dilutions of purified $His_{10}$-MBP-Orn, $His_{10}$-MBP Orn mutants, or untagged Orn in binding buffer (10 mM Tris, pH 8, 100 mM NaCl, and 5 mM $CaCl_2$) were mixed with radiolabeled nucleotides, applied to nitrocellulose sheets, dried, imaged and $K_d$ values were calculated as described previously (*Roelofs et al., 2011*; *Patel et al., 2014*).

## Aggregation assay

A colony of each strain of *P. aeruginosa* grown in LB agar plates supplemented with 50 µg/mL carbenicillin was inoculated into borosilicate glass tubes containing 2.5 mL of LB supplemented with 0.1 mM IPTG. The cultures were placed in a fly-wheel in 37˚C incubator to spin for 18 ~ 22 hr. Culture tubes were allowed to settle at room temperature for 10 min and photographed.

## Colony morphology

Indicated strains are grown overnight at 37˚C in LB agar with either 50 µg/mL of carbenicillin or 15 µg/mL of gentamicin, as appropriate. Three independent colonies are inoculated into LB with the appropriate antibiotics at 37˚C with shaking. The bacteria were subcultured and grown until $OD_{600}$ between 1.0 and 1.5. All cultures were diluted to ~10,000 CFU per mL. To observe colony size of multiple colonies of multiple strains, 10 µL of each strain were dripped in parallel on the same LB plates with the appropriate antibiotic and indicated IPTG conditions to ensure all strains were grown with the same media and conditions. For all strains tested, the control strains were always included on the same plate.

## Data deposition

The atomic coordinates and structure factors have been deposited in the Protein Data Bank, www.rcsb.org (PDB ID codes 6N6A, 6N6C, 6N6D, 6N6E, 6N6F, 6N6G, 6N6H, 6N6I, 6N6J, and 6N6K).

## Acknowledgements

This work was supported by the NIH via grant R01GM123609 (HS), R01AI110740 (VTL), R01AI142400 (to HS, WCW and VTL), Cystic Fibrosis Foundation (CF Foundation) LEE16G0 (VTL) and National Science Foundation (NSF) MCB1051440 (WCW). CAW was supported in part by the National Institutes of Health (NIH) training grant T32-AI089621. CHESS is supported by the NSF and NIH/NIGMS via NSF award DMR-1332208, and the MacCHESS resource is supported by NIH/NIGMS award GM103485.

## Additional information

### Funding

| Funder | Grant reference number | Author |
|---|---|---|
| National Institute of Allergy and Infectious Diseases | R01AI110740 | Vincent T Lee |
| National Institute of General Medical Sciences | R01GM123609 | Holger Sondermann |
| National Science Foundation | MCB1051440 | Wade C Winkler |
| Cystic Fibrosis Foundation | LEE16G0 | Vincent T Lee |
| National Institute of Diabetes and Digestive and Kidney Diseases | R01AI110740 | Vincent T Lee |
| National Institute of General Medical Sciences | T32-GM080201 | Cordelia A Weiss |
| National Institute of Allergy and Infectious Diseases | R01AI142400 | Wade C Winkler Holger Sondermann Vincent T Lee |

The funders had no role in study design, data collection and interpretation, or the decision to submit the work for publication.

### Author contributions

Soo-Kyoung Kim, Conceptualization, Formal analysis, Investigation, Writing—original draft; Justin D Lormand, Conceptualization, Formal analysis, Investigation, Writing—review and editing; Cordelia A Weiss, Conceptualization, Formal analysis, Investigation, Writing—original draft, Writing—review and editing; Karin A Eger, Husan Turdiev, Asan Turdiev, Formal analysis, Investigation; Wade C Winkler, Conceptualization, Formal analysis, Supervision, Validation, Writing—original draft, Project administration, Writing—review and editing; Holger Sondermann, Conceptualization, Formal analysis, Supervision, Funding acquisition, Validation, Investigation, Visualization, Writing—original draft, Project administration, Writing—review and editing; Vincent T Lee, Conceptualization, Formal analysis, Supervision, Funding acquisition, Validation, Writing—original draft, Project administration, Writing—review and editing

### Author ORCIDs

Holger Sondermann  https://orcid.org/0000-0003-2211-6234
Vincent T Lee  https://orcid.org/0000-0002-3593-0318

### Decision letter and Author response

Decision letter https://doi.org/10.7554/eLife.46313.sa1
Author response https://doi.org/10.7554/eLife.46313.sa2

## Additional files

### Supplementary files
- Supplementary file 1. Primers.
- Transparent reporting form

### Data availability

The atomic coordinates and structure factors have been deposited in the Protein Data Bank, www.rcsb.org (PDB ID codes 6N6A, 6N6C, 6N6D, 6N6E, 6N6F, 6N6G, 6N6H, 6N6I, 6N6J, and 6N6K). Source data files have been provided for Figures.

The following datasets were generated:

| Author(s) | Year | Dataset title | Dataset URL | Database and Identifier |
|---|---|---|---|---|
| Lormand JD, Sondermann H | 2019 | Vibrio cholerae Oligoribonuclease bound to pGG | http://www.rcsb.org/structure/6N6A | Protein Data Bank, 6N6A |
| Lormand JD, Sondermann H | 2019 | Vibrio cholerae Oligoribonuclease bound to pAA | http://www.rcsb.org/structure/6N6C | Protein Data Bank, 6N6C |
| Lormand JD, Sondermann H | 2019 | Vibrio cholerae Oligoribonuclease bound to pAG | http://www.rcsb.org/structure/6N6D | Protein Data Bank, 6N6D |
| Lormand JD, Sondermann H | 2019 | Vibrio cholerae Oligoribonuclease bound to pGA | http://www.rcsb.org/structure/6N6E | Protein Data Bank, 6N6E |
| Lormand JD, Sondermann H | 2019 | Vibrio cholerae Oligoribonuclease bound to pGC | http://www.rcsb.org/structure/6N6F | Protein Data Bank, 6N6F |
| Lormand JD, Sondermann H | 2019 | Vibrio cholerae Oligoribonuclease bound to pCG | http://www.rcsb.org/structure/6N6G | Protein Data Bank, 6N6G |
| Lormand JD, Sondermann H | 2019 | Vibrio cholerae Oligoribonuclease bound to pCpU | http://www.rcsb.org/structure/6N6H | Protein Data Bank, 6N6H |
| Lormand JD, Sondermann H | 2019 | Human REXO2 bound to pGG | http://www.rcsb.org/structure/6N6I | Protein Data Bank, 6N6I |
| Lormand JD, Sondermann H | 2019 | Human REXO2 bound to pAA | http://www.rcsb.org/structure/6N6J | Protein Data Bank, 6N6J |
| Lormand JD, Sondermann H | 2019 | Human REXO2 bound to pAG | http://www.rcsb.org/structure/6N6K | Protein Data Bank, 6N6K |

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
