## [Decision Letter]

Thank you for submitting your article "A dedicated diribonuclease resolves a key bottleneck as the terminal step of RNA degradation" for consideration by *eLife*. Your article has been reviewed by three peer reviewers, and the evaluation has been overseen by a Reviewing Editor and Gisela Storz as the Senior Editor. The following individuals involved in review of your submission have agreed to reveal their identity: Simon Dove (Reviewer #1); Ben Luisi (Reviewer #2); Arun Malhotra (Reviewer #3).

The reviewers have discussed the reviews with one another and the Reviewing Editor has drafted this decision to help you prepare a revised submission.

Summary:

The manuscript from the Lee, Sondermann, and Winkler labs describes results obtained from biochemical, genetic, and structural studies of the cellular mechanism of Oligoribonuclease (Orn), a 3'-5' exonuclease from γ-proteobacteria. The authors provide several lines of evidence indicating that Orn's cellular function is to carry out the terminal step of RNA degradation, thus serving as a "dedicated diribonuclease." These findings challenge the existing view that Orn is involved in degradation not only of diribonucleotides, but also other short RNA products up to ~5-nt in length. The substrate specificity of Orn for dinucleotides is illuminated by high resolution crystal structures of Vibrio cholerae Orn and the human homolog REXO2 bound to dinucleotide substrates. The authors also present evidence that Orn is the only diribonuclease in *Pseudomonas aeruginosa*, and that Orn's activity as a diribonuclease influences cell growth and exopolysaccharide synthesis.

Each reviewer judged the work to be significant because it challenges the existing view of Orn's cellular function as involved in the degradation of short RNA species up to ~5-nt in length. They also felt the experiments are well designed and the results are convincing, making a strong case that Orn, which is essential in some bacteria, acts at the final step of RNA degradation by digesting dinucleotides rather than larger oligonucleotide substrates.

The reviewers recommended the manuscript be accepted for publication after the text is modified to address the points raised below. In addition, each of the reviewers agreed that the title was not ideal for generating interest in the work and should be changed. One recommendation for a title that may generate more interest in the work is: "Structural basis for the terminal step of RNA degradation".

Essential revisions:

1) In the biochemical and cellular tests of the Orn mutants with substitutions at active site and phosphate cap residues (Figure 3) it is possible that the effects of some of these mutations are due to misfolding. In relation to the biochemical experiments (Figure 3A), do the authors have evidence these mutants still form dimers or are not misfolded? In relation to the cellular assays (Figure 3B), do the authors know that the mutants tested are as abundant as the wild-type protein? If additional data are not provided that address these issues, the authors should acknowledge these limitations in the text.

2) Based on data presented in Figure 5 the authors claim that Orn is the only diribonuclease in *P. aeruginosa*. The data are certainly consistent with this claim. However, it is difficult to exclude the possibility that there are other dinucleotidases that might be made under conditions that are different to those used to make the cell lysates (e.g. different media or growth phase). The data presented also cannot rule out the possibility that there may be other dinucleotidases with specific substrate specificities that are missed in their assay (only pGpG is used as substrate with lysates in Figure 5). To address these potential caveats, the authors should reword the claim in the subheading “Orn is the only diribonuclease in *P. aeruginosa*”.

3) In the subsection “Structure of Orn complexes with pGpG and other linear diribonucleotides”, please add text explaining how catalysis was prevented in the crystals.

4) Much of the prior biochemical characterization of this family of enzymes has been done with Orn from *E. coli*. The authors should comment on why *Vibrio cholerae* Orn was used for the studies presented here.

5) The binding studies and the structures lack metal ions. Orn uses divalent metal ions at its active site and this can influence substrate binding. Was EDTA added to the binding and crystallization buffers to suppress activity? Could weak residual Orn activity affect the binding analysis from the DRaCALA experiments? One approach may be to use active site mutants for the binding studies.

6) The binding assays (Table 1) also used a His-MBP-Orn fusion, while the activity and in vivo studies used non-tagged (or tag removed) Orn. Were any controls performed to ensure that MBP did not interfere with the *K_d_* measurements?

7) While the structural results show a tight binding of a dinucleotide at the Orn active-site pocket, does this really preclude activity on larger oligomers? In the electrostatic map of Orn's surface shown in Figure 2—figure supplement 1B, one prominent feature is a conserved basic patch just below the active site which could potentially help bind larger oligomers and direct them into the active site pocket. Was crystallization attempted with larger oligos?

8) The manuscript should have a more thorough discussion of other factors that may explain the low binding/activity seen here on longer oligos. Some prior studies (Cohen et al., 2015; Mechold et al., 2006) have used bulky 5'-Cy5 labeling as pointed in the manuscript. However, others (Ghosh and Deutscher, 1999; Datta and Niyogi, 1975) used substrates that were more natural. Could the discrepancy in the results be due to different buffer conditions?

9) The authors’ results would imply that there is another unknown exonuclease responsible for degrading RNA oligomers into dinucleotides. Limit products from RNase R and RNase II are larger (2-3 nts for RNR; 3-4 or more nts for RNB), as are RNA fragments generated by abortive transcription initiation. In this context, results on degradation of the 7mer even in the orn deleted cell extracts (Figure 5B) are interesting. Is it possible that the added DNase (subsection “Preparation of whole cell lysates”) in the cell extracts is responsible for this activity? Figure 7 suggests that RNA degradation gives rise to only dinucleotides and should be revised.

10) Change the heading from "The phosphate cap is required for diribonuclease activity" to "Phosphate-cap binding residues are required for diribonuclease activity." The title implies that the presence or absence of a 5' phosphate was tested, which it was not. The statement in the subsection “The phosphate cap is required for diribonuclease activity”, should also be rewritten as I do not believe that the results "demonstrate that an intact phosphate cap is required for enzyme function" as stated in the text.

---

## [Author Response]

Essential revisions:1) In the biochemical and cellular tests of the Orn mutants with substitutions at active site and phosphate cap residues (Figure 3) it is possible that the effects of some of these mutations are due to misfolding. In relation to the biochemical experiments (Figure 3A), do the authors have evidence these mutants still form dimers or are not misfolded? In relation to the cellular assays (Figure 3B), do the authors know that the mutants tested are as abundant as the wild-type protein? If additional data are not provided that address these issues, the authors should acknowledge these limitations in the text.

We have tagged wild-type and mutant alleles with HA tag and induced expression in PA14 ∆*orn* as previous done for our biological assays. Proteins were precipitated and analyzed by Western blot using anti-HA. These new data are presented in Figure 3—figure supplement 1 and suggest that all of the proteins are expressed at similar levels in PA14 ∆*orn*. In addition, we expressed and purified Orn*_Vc_* wild-type alongside point-mutants R^130^A (phosphate cap) and D^12^A (DEDD motif), and subjected proteins to size exclusion chromatography-coupled multi-angle light scattering (SEC-MALS) experiments to determine their oligomeric state. All three proteins are stable dimers as shown in Figure 3—figure supplement 2. Together, these results support our previous conclusion that mutations in these key residues ablate Orn function as a diribonuclease without affecting quaternary structure or protein expression. Discussion of these points have been added to the subsection “The phosphate cap is required for diribonuclease activity”).

2) Based on data presented in Figure 5 the authors claim that Orn is the only diribonuclease in P. aeruginosa. The data are certainly consistent with this claim. However, it is difficult to exclude the possibility that there are other dinucleotidases that might be made under conditions that are different to those used to make the cell lysates (e.g. different media or growth phase). The data presented also cannot rule out the possibility that there may be other dinucleotidases with specific substrate specificities that are missed in their assay (only pGpG is used as substrate with lysates in Figure 5). To address these potential caveats, the authors should reword the claim in the subheading “Orn is the only diribonuclease in P. aeruginosa”.

We agree that other dinucleotidases could be expressed under different conditions. To further support the statement that Orn is the only diribonuclease in *P. aeruginosa* under the assay condition, we have tested the lysates of catalytically inactive *orn* mutant alleles. We found that L^18^A and H^158^A alleles do not support the cleavage of the accumulated dinucleotide product from the degradation of the input 7-mer substrate. These results are presented in Figure 5—figure supplement 1 and indicate that Orn is the only dinucleotidase under the assay condition (subsection “Orn is the only diribonuclease in *P. aeruginosa* grown under laboratory conditions”, first paragraph). We have altered the subheading title and the text to indicate that our findings are specific for our specific assay conditions.

3) In the subsection “Structure of Orn complexes with pGpG and other linear diribonucleotides”, please add text explaining how catalysis was prevented in the crystals.

The wild-type protein appears to purify free of divalent cations. Catalysis is only observed when divalent cations are added to the reaction. As requested by the reviewer, we added a clarifying statement to the main text (subsection “Structure of Orn complexes with pGpG and other linear diribonucleotides”, first paragraph).

4) Much of the prior biochemical characterization of this family of enzymes has been done with Orn from E. coli. The authors should comment on why Vibrio cholerae Orn was used for the studies presented here.

We initially crystallized *Pseudomonas aeruginosa* Orn, consistent with the bacterial host used in this study, but the crystal packing prevented substrate binding. *Vibrio cholerae* was the next protein we tried in order to obtain alternative crystal forms that supported substrate binding. We clarified this point in text by adding the following sentence: “We initially determined the crystal structure of *P. aeruginosa* Orn (Orn*_Pa_*) with diribonucleotide substrate, however crystal packing contacts prevented substrate binding (data not shown).”.

5) The binding studies and the structures lack metal ions. Orn uses divalent metal ions at its active site and this can influence substrate binding. Was EDTA added to the binding and crystallization buffers to suppress activity? Could weak residual Orn activity affect the binding analysis from the DRaCALA experiments? One approach may be to use active site mutants for the binding studies.

See above for response to question #3. The wild-type protein appears to purify free of divalent cations, even without any special treatment (e.g. EDTA). The wild-type protein can bind substrates and crystallizes as a complex, but remains catalytically inactive. Nevertheless, we performed the suggested experiment and repeated the binding studies using the different catalytically dead mutant Orn proteins in the presence of magnesium. The catalytically inactive proteins bind with an affinity that is within an order of magnitude of wild-type in the absence of divalent cations (see Figure 3—figure supplement 1). These new results indicate that our results from our previous studies are correctly interpreted.

In addition, we initially crystallized dinucleotide-bound Orn*_Vc_* with a D^12^A mutation in the conserved DEDD motif. Structures are identical to the wild-type Orn*_Vc_* structures presented in the manuscript. We chose not to include structures obtained with active-site mutants in the manuscript since they did not add new information.

*6) The binding assays (Table 1) also used a His-MBP-Orn fusion, while the activity and* in vivo *studies used non-tagged (or tag removed) Orn. Were any controls performed to ensure that MBP did not interfere with the K_d_ measurements?*

We have tested Orn_Vc_ that was cleaved from its fusion tag (corresponding to the crystallized protein) and the *K_d_* is 80 nM. This *K_d_*value for cleaved Orn is very similar to His-MBP-Orn (90 nM). These results indicate that the His-MBP tag did not significantly alter pGpG binding affinity to Orn_Vc_ (see Figure 3—figure supplement 4B). A sentence has been added to the text to indicate that the affinity tag does not affect protein-dinucleotide affinity (subsection “Orn functions as a diribonuclease in vitro”, first paragraph).

7) While the structural results show a tight binding of a dinucleotide at the Orn active-site pocket, does this really preclude activity on larger oligomers? In the electrostatic map of Orn's surface shown in Figure 2—figure supplement 1B, one prominent feature is a conserved basic patch just below the active site which could potentially help bind larger oligomers and direct them into the active site pocket. Was crystallization attempted with larger oligos?

We also attempted crystallization of inactive Orn*_Vc_*-D^12^A with 3-, 4- and 5-mer RNAs and of wild-type protein with a 3-mer RNA. All resulting structures resolved only dinucleotides at the active site. These results are consistent with two recent reports of structural data for *Colwellia psychrerythraea* Orn and REXO2 bound to substrates (Lee et al., 2019; Chu et al., 2019). In the case of Orn, crystallization was attempted with a 5-mer RNA, however only two bases were resolved at the active site. In the report about REXO2, a similar observation was made for RNA substrates. A complex with a longer ligand was obtained with DNA, which may not allow conclusions for RNA substrates. In the same report, RNase activity against longer RNA was observed with a 20x molar excess of enzyme (1 µM) over RNA (50 nM), likely not representing physiological conditions. We discuss this work in our revised manuscript (subsection “Structure of Orn complexes with pGpG and other linear diribonucleotides”, last paragraph). Our structural analysis in combination with our biochemical data make a strong case for Orn acting as a dedicated dinucleotidase.

8) The manuscript should have a more thorough discussion of other factors that may explain the low binding/activity seen here on longer oligos. Some prior studies (Cohen et al., 2015; Mechold et al., 2006) have used bulky 5'-Cy5 labeling as pointed in the manuscript. However, others Ghosh and Deutscher, 1999; Datta and Niyogi, 1975) used substrates that were more natural. Could the discrepancy in the results be due to different buffer conditions?

The reviewer is correct to ask for a careful comparative review of these papers to provide a rationale for the vast differences between our current observations and those from previous studies. Through studying the Materials and methods sections of the publications, we found that it was challenging to compare the studies as the details within the Materials and methods sections written over a four-decade span have distinct styles and use of language that changed significantly based on the knowledge available at the time of publication. Based on our best estimation, we found large differences in the concentrations of enzyme, substrate, and enzyme to substrate ratios. Notably, we identified a consensus difference in the reaction buffer used in all other studies from the ones performed by us. In all manuscripts, the reaction buffer contains a buffer (Tris, HEPES or bis-tris propane) at pH 8.0 with 5 mM divalent cation (Mg^2+^ or Mn^2+^). However, all of the previously published studies perform Orn activity assays in reaction buffer lacking NaCl/KCl instead of the 100 mM NaCl that is used in the current manuscript and manuscripts published earlier by us. The difference in ionic strength of the reaction buffer is significant and likely contributes at least in part to the disparate results from similarly performed experiment. We argue that performing experiments in buffer with near-physiological ionic strength approximates the condition in cells. We have added two sentences to clearly indicate these differences in reaction buffer conditions (subsection “Orn functions as a diribonuclease in vitro”, last paragraph).

9) The authors’ results would imply that there is another unknown exonuclease responsible for degrading RNA oligomers into dinucleotides. Limit products from RNase R and RNase II are larger (2-3 nts for RNR; 3-4 or more nts for RNB), as are RNA fragments generated by abortive transcription initiation. In this context, results on degradation of the 7mer even in the orn deleted cell extracts (Figure 5B) are interesting. Is it possible that the added DNase (subsection “Preparation of whole cell lysates”) in the cell extracts is responsible for this activity? Figure 7 suggests that RNA degradation gives rise to only dinucleotides and should be revised.

We tested the ability of DNase, lysozyme and PMSF individually or in combination to hydrolyze the 7-mer RNA. Our results indicate that none of these additives contributed to the hydrolysis of 7 base oligonucleotide (see Author response image 1). Since this is not particularly relevant to the manuscript, we opt to present the data only in the Response to Reviewer document.

**Author response image 1. respfig1:** Additives to cell lystaes do not contribute to cleavage of ^32^P-AAAAAGG oligoribonucleotide. 10 µg/ml DNase, 25 µg/ml lysozyme, 1 mM PMSF, and the combination of all three additives were test for cleavage of ^32^P-AAAAAGG. The samples were stopped at indicated times and analyzed by 20% Urea gel. All data shown represent duplicate independent experiments.

The identity of enzyme(s) responsible for degrading the 3-4 mer RNA fragments to dinucleotides is currently unknown. Future studies will be required to identify these enzymes. We modified the model in Figure 7 as requested.

10) Change the heading from "The phosphate cap is required for diribonuclease activity" to "Phosphate-cap binding residues are required for diribonuclease activity." The title implies that the presence or absence of a 5' phosphate was tested, which it was not. The statement in the subsection “The phosphate cap is required for diribonuclease activity”, should also be rewritten as I do not believe that the results "demonstrate that an intact phosphate cap is required for enzyme function" as stated in the text.

We included new data in the amended manuscript showing that GpG added in vast (1000x) excess over radiolabed pGpG has no effect on pGpG binding (or cleavage), whereas 1000x unlabeled pGpG abolishes radioligand binding completely (Figure 2—figure supplement 4). Attempts to co-crystallize Orn*_Vc_* with GpG were also unsuccessful so far, further suggesting that GpG has significantly lower affinity to Orn than pGpG. With this, we argue that the chosen subheading accurately describes our data. Reference to the new figure supplement can be found in the first paragraph of the subsection “Interaction of Orn with substrates”.